# PINP: Physics-Informed Neural Predictor with Latent Estimation of Fluid Flows

**Huaguan Chen[1], Yang Liu[2], Hao Sun[1],***
[1]Gaoling School of Artificial Intelligence, Renmin University of China, Beijing, China
[2]School of Engineering Science, University of Chinese Academy of Sciences, Beijing, China
Emails: `huaguanchen@ruc.edu.cn`; `liuyang22@ucas.ac.cn`; `haosun@ruc.edu.cn`

## Abstract

Accurately predicting fluid dynamics and evolution has been a long-standing challenge in physical sciences. Conventional deep learning methods often rely on the nonlinear modeling capabilities of neural networks to establish mappings between past and future states, overlooking the fluid dynamics, or only modeling the velocity field, neglecting the coupling of multiple physical quantities. In this paper, we propose a new physics-informed learning approach that incorporates coupled physical quantities into the prediction process to assist with forecasting. Central to our method lies in the discretization of physical equations, which are directly integrated into the model architecture and loss function. This integration enables the model to provide robust, long-term future predictions. By incorporating physical equations, our model demonstrates temporal extrapolation and spatial generalization capabilities. Experimental results show that our approach achieves the state-of-the-art performance in spatiotemporal prediction across both numerical simulations and real-world extreme-precipitation nowcasting benchmarks.

## 1 Instruction

The prediction and analysis of fluid dynamics play a crucial role across many fields of science. Fluid phenomena span a wide range of scales, from macroscopic atmospheric circulation to microscopic intracellular transport, underscoring the complexity and significance of fluid motion. However, noisy data, experimental limitations, and the inaccessibility of physical quantities present substantial challenges in learning the underlying dynamical systems and accurately predicting future fluid evolution.

These challenges primarily arise for two reasons. First, latent physical quantities are inherently difficult to obtain. For example, the advection and diffusion of substances within fluid flows are governed by the Navier-Stokes (NS) equations, which couple multiple variables—such as velocity, pressure, and concentration—into a complex, interdependent system. Capturing real-time data, like velocity and pressure, without interference is highly challenging. Successful approaches usually rely on supervised learning for these quantities, but such data is not easy to obtain in practical scenarios. While the HFM (Raissi et al., 2020) has made notable progress by inferring latent physical quantities from PDEs and observed concentration data, it still falls short of predicting future fluid evolution.

Second, the complexity of the NS equations presents a substantial obstacle. These nonlinear equations couple multiple physical quantities, making the direct prediction of future fluid behavior nearly impossible. A promising approach is the use of neural operators, which predict future physical fields based on past observations without explicitly solving the NS equations (Li et al., 2020; Tran et al., 2023; Wang et al., 2024). These models leverage the powerful nonlinear approximation capabilities of neural networks to map past fluid states to future. However, the absence of physical constraints often limits their interpretability and generalization ability. Another approach involves approximating and simplifying the NS equations through methods like kernel methods or other theorems and applying neural networks to estimate the fluid velocity field. By combining the estimated velocity field with simplified equations, these methods attempt to predict future fluid behavior (Deng et al., 2023; Xing et al., 2024). However, these methods lack intuitive clarity, and focusing solely on the velocity field overlooks the complex multivariable coupling intrinsic to the NS equations.

---

*Corresponding author

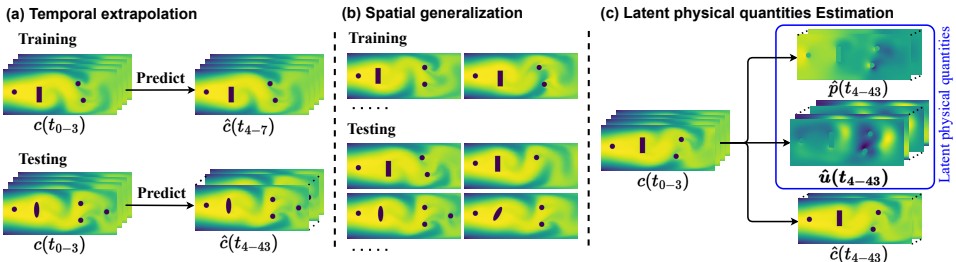

Figure 1: Model has the following capabilities: (a) Temporal extrapolation, (b) Spatial generalization, (c) Latent physical quantities Estimation.

In this paper, we propose the Physics-Informed Neural Predictor, a new approach designed to address the challenges outlined above. For fluid dynamics prediction, we discretize the NS equations in both time and space, incorporating them into model architecture and loss functions. Neural networks are employed to establish a mapping function from past measurement (observed data) to both intermediate observed quantities and latent physical variables, enabling predictions through the discretized NS equations. Unlike methods that solely establish mappings between past and future states or focus only on velocity modeling, our approach estimates multiple physical quantities (e.g., velocity and pressure) to assist in the prediction. To ensure the physical validity of the estimated physical quantities, we impose both physical and temporal constraints. By explicitly modeling physical quantities and incorporating the NS equations into the model, our method inherently possesses interpretability and enhances the extraction of latent fluid dynamics from observed data. This integration further improves the model's temporal extrapolation and spatial generalization capabilities in prediction.

To evaluate the effectiveness of the proposed method, we conducted a comparative study against several state-of-the-art models for predicting future observed data. The evaluation was performed using both numerically simulated 2D and 3D datasets, as well as real-world nowcasting benchmarks. The predictive accuracy of the models was quantitatively assessed across various scenarios.

Experimental results demonstrate that our approach achieves SOAT performance in predicting observed data, while simultaneously estimating multiple latent physical quantities for interpretability, and exhibiting better temporal extrapolation and spatial generalization capabilities (Figure 1).

In summary, our main technical contributions align to tackle the key challenges, given by:

1) **Estimation of Multiple Physical Quantities and Assisted Prediction**: We estimate latent physical quantities, which can assist in the prediction of observable physical quantities and serve as interpretable evidence to support our predictions.

2) **Temporal Extrapolation and Spatial Generalization**: Our model demonstrates the ability to extrapolate beyond the training steps and generalize across varying spatial domains.

3) **Superior Performance**: Our model consistently achieves SOTA performance across a wide array of benchmark tests, including synthetic and real-world datasets (both 2D and 3D).

## 2 RELATED WORK

Modeling fluid flows has always been a significant challenge. Since it is typically impossible to calculate explicit solutions for NS equations, scientists have developed numerous numerical methods to address this problem, such as the finite element method (Donea & Huerta, 2003), finite difference method (Godunov & Bohachevsky, 1959), finite volume method (Jasak, 1996), and spectral method (Orszag, 1979). However, these numerical methods heavily rely on initial conditions, which are often difficult to obtain in practical applications. Moreover, numerical methods face high computational costs, especially when dealing with complex and variable scenarios and large computational domains. The complexity of modeling and the lengthy computation times render numerical methods impractical in such cases. In recent years, the development of deep learning technologies has provided new possibilities for solving this problem. Based on whether latent physical quantities are modeled, we can broadly categorize these approaches into the following methods.

**Neural Operator methods**. This category of methods primarily leverages the powerful nonlinear modeling capabilities of neural networks to approximate the complex mapping between inputs and

outputs, thereby enabling predictions based on past fluid data(Rao et al., 2023). For example, Deep-ONet (Lu et al., 2021a), which is derived from the universal approximation theorem, employs a branch-trunk architecture. FNO (Li et al., 2020) approximates the integral operator by employing linear transformations in the Fourier space. U-FNO (Wen et al., 2022) and U-NO (Rahman et al., 2022) enhance FNO with U-shaped multi-scale frameworks. Geo-FNO (Li et al., 2023b) handles complex geometries by transforming irregular input physical domains into latent spaces with uniform grids. F-FNO (Tran et al., 2023) improves FNO through separable spectral layers and residual connections. Additionally, MWT (Gupta et al., 2021) introduces a multiwavelet-based neural operator. LSM (Wu et al., 2023) decomposes complex nonlinear operators into multiple base operators, using neural spectral blocks to solve high-dimensional PDEs. However, these methods do not explicitly model latent physical quantities, leading to a lack of physical interpretability. Overlooking the fluid dynamics, they perform poorly in temporal extrapolation and spatial generalization.

**Latent physical quantity modeling methods**. This category of methods is based on existing NS equations and explicitly models the required physical quantities, such as velocity, for fluid prediction. One approach is to directly incorporate the NS solutions as parameters of the deep model and formalize the constraints, e.g., PDEs, corresponding initial and observed data, as a loss function (Raissi et al., 2019; 2020; Lu et al., 2021b). enabling the inference of latent physical quantities from observed data. However, this method is not effective for predicting future fluid behavior. Another approach is to approximate and simplify NS equations and estimate the fluid velocity field, then use the velocity field and the simplified equations to predict future fluid behavior. For instance, Zhang et al. (2022) initially provided estimates using an optical flow predictor constrained by the NS equations. DVP (Deng et al., 2023) combines vortex particles with a vortex-to-velocity dynamics mapping to capture complex flow dynamics. HelmFluid (Xing et al., 2024) integrates learned Helmholtz dynamics to generate future fluid behavior. However, this method does not achieve explicit incorporating of the NS equations, lacking intuitiveness. In addition, since fluid dynamics is a complex system involving the coupling of multiple physical quantities, models that rely solely on the velocity field have inherent limitations.

## 3 PHYSICS-INFORMED NEURAL PREDICTOR

Let us consider a scenario: the transport of dye in water. The dye is transported and diffused with the flow of water, and it does not affect the flow of the water. Under normal temperature and pressure, water approximately satisfies the incompressible Navier-Stokes equations. We aim to predict the future evolution of the dye concentration only based on observed dye concentration data, while also estimating the unobserved velocity and pressure fields of the water as interpretable evidence.

Figure 2 illustrates the overall architecture of our PINP model. Herein, the dye flows from left to right through a pipe with obstacles. The observed data includes dye concentration $c$ and spatial information $\psi$. Using these observed data, and based on a physics-informed neural network, we simultaneously estimate the concentration field at time $t'$, as well as the underlying velocity and pressure fields which are unmeasured. With these inferred fields and the Discrete PDEs Prediction Network, we can predict the concentration field $\hat{c}'$ for the next time step.

In the Discrete PDEs Prediction Network, we construct the predictor using discretized PDEs. Since this discretization naturally introduces numerical errors, we use a correction network to refine the predicted concentration field, resulting in the final prediction $\hat{c}'$. Additionally, the simultaneously estimated velocity and pressure fields at time $t'$ serve as interpretable evidence.

**Notation**. Throughout the paper, $\nabla = \nabla_{\mathbf{x}}$ denotes spatial gradients, $\dot{c} = \frac{dc}{dt}$ time derivatives, $\nabla \cdot \boldsymbol{u} = \mathrm{tr}(\nabla \boldsymbol{u})$ divergence, and $\cdot$ tensor inner product. $k \in \{1, 2, ..., N\}$ denotes the $k$-th frame, $t_k$ the time of $k$-th frame, and $\Delta t$ the time interval between $t_k$ and $t_{k+1}$.

### 3.1 DISCRETIZATION PDES

We considered the transport of a passive scalar $c(\boldsymbol{x}, t) \in \mathbb{R}^{H \times W}$ by a flow velocity field $\boldsymbol{u}(\boldsymbol{x}, t) \in \mathbb{R}^{2 \times H \times W}$. The passive scalar is advected by the flow and diffused but has no dynamical effect on the fluid motion itself, which satisfies the incompressible NS equations:

$$\dot{c}(\boldsymbol{x}, t) = -\boldsymbol{u}(\boldsymbol{x}, t) \cdot \nabla c(\boldsymbol{x}, t) + \mathrm{Pe}^{-1} \nabla^2 c(\boldsymbol{x}, t), \tag{1}$$

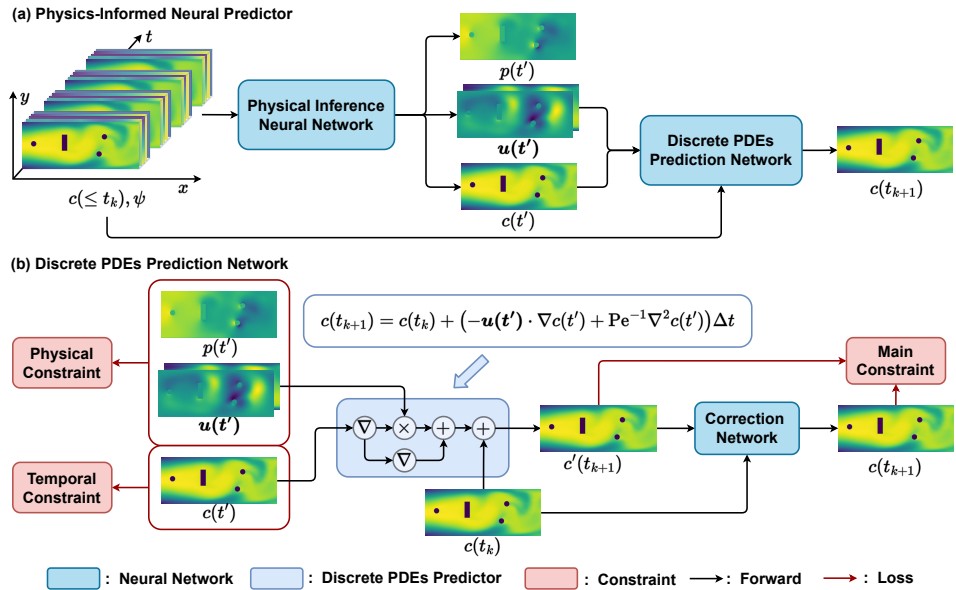

Figure 2: Schematic architecture of the proposed Physics-Informed Neural Predictor (PINP).

where Pe denotes the Peclét number, $c(\boldsymbol{x}, t)$ the concentration of the passive scalar , which can be observed without interfering with the state of fluid flow.

According to the aforementioned equations, if we need to predict the concentration at future time points, we can derive the expression for the future concentration as follows:

$$c(\boldsymbol{x}, t_{k+1}) = \int_{t_k}^{t_{k+1}} \dot{c}(\boldsymbol{x}, t) dt + c(\boldsymbol{x}, t_k). \qquad (2)$$

The integration term in Eq. 2 is undoubtedly extremely complex and challenging to solve. However, considering the continuity of material concentration $c(\boldsymbol{x}, t)$ over time, based on the Lagrange Mean Value Theorem, there must exist a moment $t'$ between time $t_k$ and $t_{k+1}$ such that:

$$\int_{t_k}^{t_{k+1}} \dot{c}(\boldsymbol{x}, t) dt = c(\boldsymbol{x}, t')\Delta t, \ t' \in [t_k, t_{k+1}]. \qquad (3)$$

Therefore, the prediction process can be described by the following equation:

$$\begin{aligned} c(\boldsymbol{x}, t_{k+1}) &= c(\boldsymbol{x}, t_k) + c(\boldsymbol{x}, t')\Delta t \\ &= c(\boldsymbol{x}, t_k) + \left(-\boldsymbol{u}(\boldsymbol{x}, t') \cdot \nabla c(\boldsymbol{x}, t') + \text{Pe}^{-1}\nabla^2 c(\boldsymbol{x}, t')\right)\Delta t. \end{aligned} \qquad (4)$$

Eq. 4 is the final discrete form of the NS equations that we utilize.

## 3.2 Physical Inference Neural Network

Based on Eq. 4, we can predict the concentration. We need a way to model concentration data $c(\boldsymbol{x}, t')$, velocity data $\boldsymbol{u}(\boldsymbol{x}, t')$, and the Peclét number (Pe) included in the equation. Considering the powerful nonlinear fitting capabilities of neural networks, we utilize them to establish the mapping function from past $K$ frames of data to the physical quantities at time $t'$:

$$\phi(\boldsymbol{x}, t') = \mathcal{F}_{\theta_\phi}(\boldsymbol{c}(\boldsymbol{x}, \le t_k), \psi), \ t' \in [t_k, t_{k+1}], \qquad (5)$$

where $\phi$ denotes the physical quantities, $\boldsymbol{c}(\boldsymbol{x}, \le t_k) = [c(\boldsymbol{x}, t_{k+1-K}), ..., c(\boldsymbol{x}, t_{k-1}), c(\boldsymbol{x}, t_k)] \in \mathbb{R}^{K \times H \times W}$, $\psi \in \mathbb{R}^{C \times H \times W}$ is the spatio embeddings.

For spatial information $\psi$, we define it as $\psi = (\boldsymbol{x}, d, b)$, where $\boldsymbol{x} \in \mathbb{R}^{2 \times H \times W}$ denotes the coordinates of grid points. For bounded fluids, $d$ and $b$ collectively characterize the internal positions and boundary information of the space. $d \in \mathbb{R}^{H \times W}$ denotes the Signed Distance Function (SDF, Osher & Sethian, 1988), which describes the shortest distance from each position in the physical field to

the obstacles. $b \in \mathbb{R}^{H \times W}$ is used to describe various attributes of each position in the physical field, with different values of $b$ representing boundaries and regions that permit fluid flow.

We set Pe as a learnable parameter in the network since the Pe is a constant.

### 3.3 DISCRETE PDES PREDICTION NETWORK

Based on Eq. 4, we define the Discrete PDEs Predictor:

$$\hat{c}'(\boldsymbol{x}, t_{k+1}) = c(\boldsymbol{x}, t_k) + \left(-\boldsymbol{u}(\boldsymbol{x}, t') \cdot \nabla c(\boldsymbol{x}, t') + \mathrm{Pe}^{-1} \nabla^2 c(\boldsymbol{x}, t')\right) \Delta t. \tag{6}$$

Since the discretized PDE predictor may introduce errors, the predicted concentration field is corrected using a correction neural network ($\mathcal{G}_\theta$):

$$\hat{c}(\boldsymbol{x}, t_{k+1}) = \mathcal{G}_\theta(\hat{c}'(\boldsymbol{x}, t_{k+1}), \hat{c}(\boldsymbol{x}, t_k)). \tag{7}$$

### 3.4 SPECIFIC MODEL IMPLEMENTATION

In Eq. 5, the mapping function we need to establish is a complex nonlinear function with coupled multiple physical quantities and multi-scale features. Since U-Net (Ronneberger et al., 2015) is well-suited for modeling multi-scale information, 3D U-Net is considered more effective in capturing spatiotemporal features. Here, we adopt 3D U-Net (Çiçek et al., 2016) for the mapping:

$$\mathcal{F}_{\theta_\phi}(\boldsymbol{c}(\boldsymbol{x}, \leq t_k), \psi) = \mathcal{F}_{\text{3D U-Net}}(\boldsymbol{c}(\boldsymbol{x}, \leq t_k), \psi). \tag{8}$$

Similarly, in Eq. 7, we establish our correction network using the U-Net architecture. Since the correction task is relatively simple, the model used here is more lightweight. Appendix A presents more details, including the gradient operator, signed distance field, mask settings and model structure.

$$\mathcal{G}_\theta(\hat{c}'(\boldsymbol{x}, t_{k+1}), \hat{c}(\boldsymbol{x}, t)) = \mathcal{G}_{\text{U-Net (lightweight)}}(\hat{c}'(\boldsymbol{x}, t_{k+1}), \hat{c}(\boldsymbol{x}, t_k)). \tag{9}$$

### 3.5 LOSS FUNCTION

Our loss function consists of three components: Data, Physical, and Temporal constraints. These components ensure the prediction accuracy and inference of underlying physical quantities.

**Data constraints**. Using the observed data (training labels), we impose constraints on the predicted values both before and after correction:

$$\mathcal{L}_{\text{Data}}(\boldsymbol{x}, t_k) = ||\hat{c}'(\boldsymbol{x}, t_k) - c(\boldsymbol{x}, t_k)||_2^2 + ||\hat{c}(\boldsymbol{x}, t_k) - c(\boldsymbol{x}, t_k)||_2^2. \tag{10}$$

**Physical constraints**. We consider the physical constraints of the incompressible NS equations:

$$\dot{\boldsymbol{u}}(\boldsymbol{x}, t) = -\boldsymbol{u}(\boldsymbol{x}, t) \cdot \nabla \boldsymbol{u}(\boldsymbol{x}, t) - \nabla p(\boldsymbol{x}, t) + \mathrm{Re}^{-1} \nabla^2 \boldsymbol{u}(\boldsymbol{x}, t),$$
$$\nabla \cdot \boldsymbol{u}(\boldsymbol{x}, t) = 0. \tag{11}$$

Over a very short time interval, we assume that $\boldsymbol{u}(\boldsymbol{x}, t') \approx \boldsymbol{u}(\boldsymbol{x}, t_k)$ and $p(\boldsymbol{x}, t') \approx p(\boldsymbol{x}, t_k)$. We establish the residual terms by discretizing and non-dimensionalizing the aforementioned equations:

$$\boldsymbol{e_1}(\boldsymbol{x}, t_k) = \Delta \boldsymbol{u}(\boldsymbol{x}, t_k) + \left(-\boldsymbol{u}(\boldsymbol{x}, t_k) \cdot \nabla \boldsymbol{u}(\boldsymbol{x}, t_k) - \nabla p(\boldsymbol{x}, t_k) + \mathrm{Re}^{-1} \nabla^2 \boldsymbol{u}(\boldsymbol{x}, t_k)\right) \Delta t,$$
$$e_2(\boldsymbol{x}, t_k) = (\nabla \cdot \boldsymbol{u}(\boldsymbol{x}, t_k)) \Delta t. \tag{12}$$

So, we define the Physical constraint as:

$$\mathcal{L}_{\text{Physical}}(\boldsymbol{x}, t_k) = ||\boldsymbol{e_1}(\boldsymbol{x}, t_k)||_2^2 + ||e_2(\boldsymbol{x}, t_k)||_2^2. \tag{13}$$

Notably, in Eq. 12, we have introduced the pressure term $p(\boldsymbol{x}, t') \in \mathbb{R}^{H \times W}$ and the Reynolds number Re. Similar to our approach for handling concentration and velocity, we utilize neural networks to establish the mapping from the past observed data to the pressure. Re is treated as a learnable parameter. In this way, we introduce $H \times W + 1$ new variables while adding $2 \times H \times W$ constraint equations, further enhancing the physical constraints.

**Temporal constraints**. In Eq. 3, we have $t' \in [t_k, t_{k+1}]$. However, directly imposing constraints on $t'$ is both challenging and unnecessary. Instead, we applying constraints directly on $c(\boldsymbol{x}, t')$. Over a

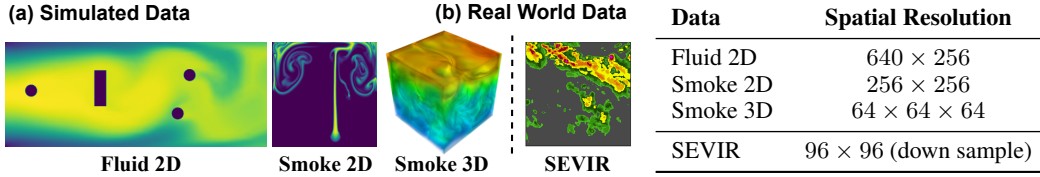

Figure 3: Overview of the data with spatial resolution.

very short time interval, assuming steady flow, we can approximate the temporal evolution of $c(\boldsymbol{x}, t)$ as a monotonic bounded function. Since $t'$ lies between $t_k$ and $t_{k+1}$, we have:

$$||c(\boldsymbol{x}, t') - c(\boldsymbol{x}, t_k)||_2^2 \leq ||c(\boldsymbol{x}, t_{k+1}) - c(\boldsymbol{x}, t_k)||_2^2. \tag{14}$$

Based on this, we establish the residual term as follows:

$$\mathcal{L}_{\text{Temporal}}(\boldsymbol{x}, t_k) = \max\{||c(\boldsymbol{x}, t') - c(\boldsymbol{x}, t_k)||_2^2 - ||c(\boldsymbol{x}, t_{k+1}) - c(\boldsymbol{x}, t_k)||_2^2, 0\}. \tag{15}$$

This implies that if $||c(\boldsymbol{x}, t') - c(\boldsymbol{x}, t_k)||_2^2$ is less than or equal to $||c(\boldsymbol{x}, t_{k+1}) - c(\boldsymbol{x}, t_k)||_2^2$, we consider the concentration at time $t'$ to be optimized within an acceptable error range, and the loss term for this component is set to 0. However, if $||c(\boldsymbol{x}, t') - c(\boldsymbol{x}, t_k)||_2^2$ is greater than $||c(\boldsymbol{x}, t_{k+1}) - c(\boldsymbol{x}, t_k)||_2^2$, we consider the concentration value at time $t'$ to require further optimization.

In summary, our loss function assembles the above constraints and is written as:

$$\mathcal{L} = \frac{1}{NHW} \sum_{i=1}^{N} \left( \mathcal{L}_{\text{Data}}(\boldsymbol{x}, t_k) + \mathcal{L}_{\text{Physical}}(\boldsymbol{x}, t_k) + \mathcal{L}_{\text{Temporal}}(\boldsymbol{x}, t_k) \right). \tag{16}$$

## 4 EXPERIMENTS

We extensively evaluate our model across four benchmarks, including both simulated and real-world scenarios, as well as 2D and 3D settings. The training and testing data consist solely of observable data. Overview of the data with spatial resolution can be found in Figure 3.

### 4.1 SIMULATED DATA

**Baselines.** Given the simulated data, we compare our model with seven recognized advanced models, including five baselines that are purely data-driven to approximate complex mapping: FNO (Li et al., 2020), F-FNO (Tran et al., 2023), U-FNO (Wen et al., 2022), U-NO (Rahman et al., 2022) and LSM (Wu et al., 2023), and a baseline that utilizes established velocity fields to assist predictions: HelmFluid (Xing et al., 2024), and two other baselines for vision tasks: U-Net (Ronneberger et al., 2015) and ResNet (He et al., 2016).

Table 1: Comparison of prediction performance on synthetic data with test metrics of MSE (↓) and MAE (↓). For clarity, the best result is shown in **bold** and the second best is underlined.

| Method | Flow 2D | | Smoke 2D | | Smoke 3D | |
|---|---|---|---|---|---|---|
| | MAE | MSE | MAE | MSE | MAE | MSE |
| U-Net (2015) | 0.0362 | 0.0076 | 0.0346 | 0.0140 | 0.1397 | 0.0385 |
| ResNet (2016) | 0.0424 | 0.0092 | 0.0412 | 0.0192 | 0.1460 | 0.0419 |
| FNO (2020) | 0.0300 | 0.0029 | 0.0303 | 0.0118 | 0.1465 | 0.0413 |
| U-NO (2022) | 0.0183 | 0.0008 | 0.0284 | 0.0092 | 0.1386 | 0.0363 |
| U-FNO (2022) | 0.0187 | 0.0011 | 0.0335 | 0.0125 | 0.1237 | 0.0314 |
| F-FNO (2023) | 0.0336 | 0.0055 | 0.0282 | 0.0107 | 0.1124 | 0.0250 |
| LSM (2023) | 0.0338 | 0.0037 | 0.0311 | 0.0109 | 0.1193 | 0.0293 |
| HelmFluid (2024) | 0.1222 | 0.0503 | 0.0254 | 0.0085 | 0.1217 | 0.0298 |
| PINP (this work) | **0.0107** | **0.0003** | **0.0209** | **0.0057** | **0.1073** | **0.0249** |

**Fluid 2D**. We consider a scenario where a substance is transported by fluid flow through a pipe. An incompressible fluid with a certain concentration flows from left to right, passing through multiple solid objects with varying positions and shapes. The substance is advected and diffuses with the fluid, and its concentration is the observed data. The real-world scenario involves the transport of dye carried by water flow through a pipe. To evaluate the model's temporal extrapolation and spatial generalization capabilities, the training process involves predicting the next 5 frames of concentration based on the previous 4 frames. During testing, the model is challenged to predict 40 future frames of concentration from the 4-frame input. Additionally, the test set includes variations such as the addition, removal and shape changes of obstacles, which were not present in training set.

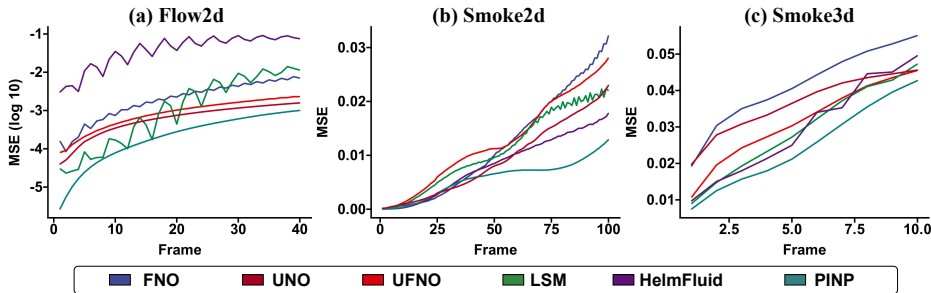

Figure 4: Comparison of MSE for different models on each prediction frame.

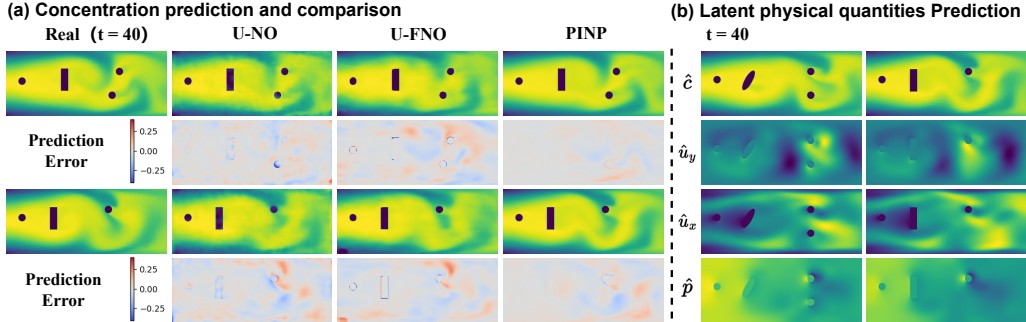

Figure 5: (a) A comparison of the 40th frame prediction results between our method and U-NO, U-FNO. (b) The inferred and predicted velocity and pressure fields at the 40th frame.

As shown in Table 1, PINP significantly outperforms other models, achieving a notable reduction in error compared to the second-best model (MAE: 0.0107 vs. 0.0183). In the frame-by-frame fluid prediction comparison (Figure 4 (a)), our model also outperforms others.

To further illustrate the effectiveness of our model, Figure 5 (a) presents several examples. Compared to U-NO and U-FNO, our model predicts fluid evolution more accurately, particularly in regions where the substance interacts with boundaries. Additionally, our model estimates velocity and pressure fields at each prediction step, providing interpretable evidence that supports the predictions and highlights its advantages in capturing complex dynamics. Figure 5 (b) shows the estimated latent physical quantities at the 40th predicted frame. The inferred physical quantities have physical significance, including the velocity distribution at the edges of obstacles and the pressure changes before and after the obstacles. These quantities can serve as interpretable evidence for predictions.

In Figure 4 (a), some of the baseline models exhibit periodic oscillations in the MSE as the number of prediction frames increases. This phenomenon is quite unusual because, under normal circumstances, a well-trained model should not exhibit such erratic oscillations. This might be due to rollout training with a small number of steps. We have discussed this issue in Supplementary Material.

**Smoke 2D**. Smoke rises and diffuses in an enclosed space, with smoke concentration as the observable data. During training, the model predicts the next 10 frames of concentration based on the previous 4 frames. During testing, the model need to predict 100 future frames based on the first 4 frames input, with variations in smoke source locations between the training and testing sets.

In the 2D smoke concentration prediction task, PINP significantly outperforms other models (Table 1). In the frame-by-frame prediction comparison (Figure 4 (b)), both HelmFluid and U-NO initially perform better than our model. But their prediction errors increase rapidly as the prediction progresses. This demonstrates the temporal extrapolation capability of our model.

Figure 6 (a) presents an example showing that, compared to U-NO and HelmFluid, our model predicts the movement of smoke more accurately, particularly the shape of the smoke plume and the direction of the smoke column. Additionally, Figure 6 (b) shows a comparison between our inferred and predicted latent physical quantities and the ground truth in the smoke region. The velocity and pressure fields, estimated solely from observable data, resemble the true values, effectively serving as evidence for the interpretability of our method.

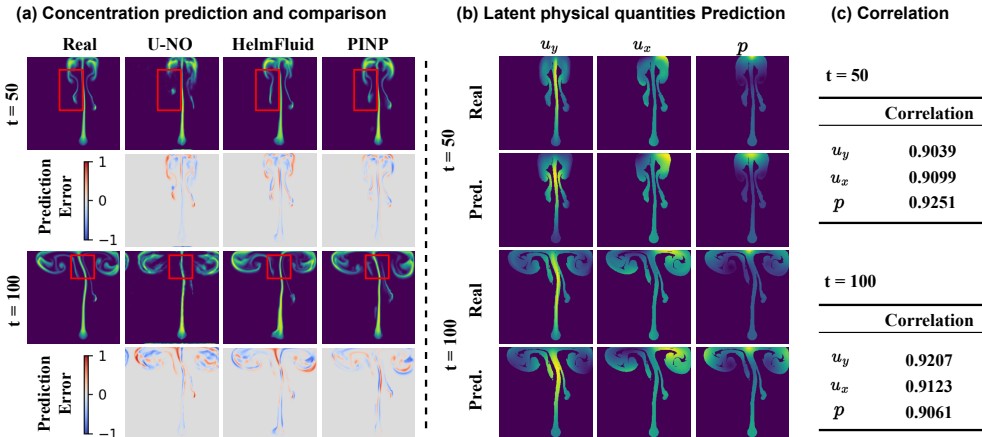

Figure 6: (a) A comparison of prediction results between our method and U-NO, Helmfluid. (b) A comparison between our predicted latent physical quantities and the ground truth and (c) Correlation.

As shown in Figure 6 (c), a high correlation ($\geq 0.9$) demonstrates the similarity of our estimated latent physical quantities with the true data, indicating that the estimated physical quantities contribute to understanding the future motion process. However, it is important to note that, due to the unknown physical scale, a scaling relationship exists between the estimated latent physical quantities and their true values. To obtain the true values of these estimated quantities, the initial values of the real physical quantities or any real data point are necessary to determine the scaling constant. A detailed discussion of this is provided in Appendix D.

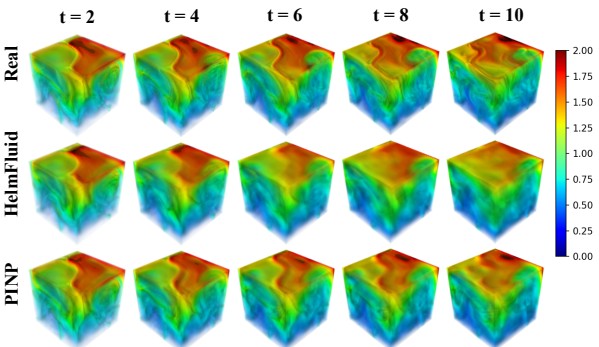

Figure 7: A comparison between our method and Helmfluid for 3D smoke prediction, with the predicted frames increasing from left to right.

**Smoke 3D**. Similar to the 2D smoke scenario, we extend our method to a 3D setting. During training, the model predicts the future 4 frames of concentration based on the past 4 frames. During testing, the model is required to predict the future 10 frames of concentration using the past 4 frames, with varying smoke source locations and concentrations between the training and testing sets.

In the 3D smoke concentration prediction task, PINP significantly outperforms other models (Table 1). In the frame-by-frame prediction comparison (Figure 4 (c)), our model also outperforms others. Figure 7 presents an example. Compared to Helmfluid, our model preserves more detailed features. The prediction results on the top surface are significantly better than those predicted by Helmfluid.

## 4.2 REAL-WORLD DATA

We consider the nowcasting prediction problem in real scenarios, which can be viewed as the transport problem of water-bearing clouds driven by wind. Unlike the simulated data, nowcasting data is boundary-less and contains noise. It is important to note that the baseline from the simulated data test is not applied to this task, and we establish new baselines for this problem.

**Baselines**. The baselines for nowcasting include video prediction models: ConvLSTM (Shi et al., 2015), PredRNN (Wang et al., 2022), SimVP (Gao et al., 2022a) and Earthformer (Gao et al., 2022b). And a physics-driven prediction model: NowcastNet (Zhang et al., 2023). NowcastNet uses a deterministic prediction model (Evolution network) paired with a probabilistic model (GAN). The Evolution network outperforms the GAN-based model in most metrics. Results for the Evolution network are shown in Table 2, while full comparisons of NowcastNet are in Appendix A (Table S1).

Table 2: Comparison of prediction performance on SEVIR. The test metrics are MSE(↓) and CSI(↑). For clarity, the best result is shown in **bold** and the second best result is underlined.

| Method | MSE (1e-3) | CSI-M | CSI-219 | CSI-181 | CSI-160 | CSI-133 | CSI-74 | CSI-16 |
|---|---|---|---|---|---|---|---|---|
| ConvLSTM (2015) | 4.0082 | 0.3490 | 0.0340 | 0.1441 | 0.2003 | 0.3274 | 0.6574 | 0.7293 |
| PredRNN (2022) | 4.0383 | 0.3382 | 0.0155 | 0.1203 | 0.1880 | 0.3252 | 0.6551 | 0.7283 |
| SimVP (2022a) | **3.8508** | 0.3567 | 0.0709 | 0.1496 | 0.1997 | 0.3212 | 0.6562 | **0.7424** |
| Earthformer (2022b) | 4.4029 | 0.3585 | 0.0661 | 0.1663 | 0.2163 | 0.3400 | 0.6428 | 0.7196 |
| NowcastNet (2023) | 3.8883 | 0.3743 | 0.0803 | 0.1811 | 0.2351 | 0.3515 | **0.6635** | 0.7339 |
| PINP (this work) | 4.1684 | **0.3800** | **0.0989** | **0.1952** | **0.2448** | **0.3573** | 0.6556 | 0.7279 |

**SEVIR**. The Storm EVent ImageRy (SE-VIR) dataset (Veillette et al., 2020) contains meteorological events across the United States between 2017 and 2019. For nowcasting, we use the NEXRAD radar composite of Vertically Integrated Liquid (VIL), which covers an area of 384 km × 384 km with a spatial resolution of 1 km and a temporal interval of 5 minutes. Following Gao et al. (2022b), we evaluate nowcasting by predicting VIL for up to 60 minutes (12 frames) based on 65 minutes of past VIL data (13 frames). We downsample the resolution to 96 × 96.

As shown in Table 2, while our model does not lead in MSE or low CSI metrics, it achieves the best results in the average CSI and high CSI metrics, highlighting its potential in extreme precipitation prediction. Figure 8 presents an comparison of our model against other baselines. In nowcasting, the NS equations are not strictly satisfied. This test evaluates the model's predictive performance and robustness to noise under conditions where these equations are not fully obeyed.

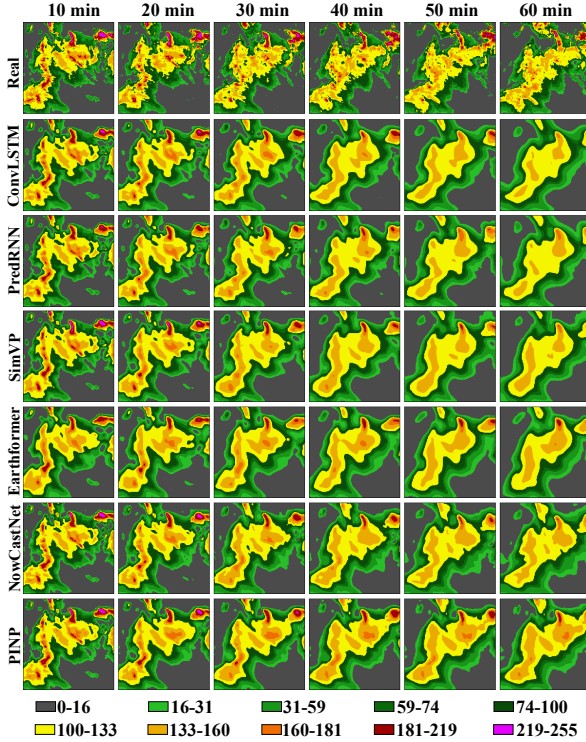

Figure 8: Comparison of results on the SEVIR dataset.

Appendix B presents more details, including the generation methods, data scale, details of SEVIR and the evaluation metrics for the test results. Details regarding the training setup, hyperparameter configuration, and multi-loss function training can be found in Appendix C. Further baseline results and physical field inference results can be found in Appendix F.

## 5 ABLATION STUDIES

We conduct ablation experiments on constraints, model components, and the discrete PDEs Predictor. Results are shown in Table 3, while Figure 9 illustrates the change in MSE with the frame under different ablation experiments. The legends (e.g., C1, M1, D1) in Figure 9 correspond to the ablation case labels in Table 3. Details for these ablation experiments can be found in Appendix E.

Table 3: Ablation Experiment Setup and Results

| Ablation Studies Setting | | MAE | MSE |
|---|---|---|---|
| **Constraint** | C1. Normal | 0.0209 | 0.0057 |
| | C2. no Physical Constraint | 0.0272 | 0.0089 |
| | C3. no Temporal Constraint | 0.0293 | 0.0108 |
| | C4. no Velocity-Pressure Constraint | 0.0262 | 0.0087 |
| **Model** | M1. Normal | 0.0209 | 0.0057 |
| | M2. no Correction Network | 0.0316 | 0.0098 |
| **Discrete PDEs** | D1. Normal | 0.0107 | 0.0003 |
| | D2. changing Discrete PDEs | 0.0197 | 0.0010 |
| | D3. replacing $c(t')$ with $c(t_k)$ | Inf | Inf |

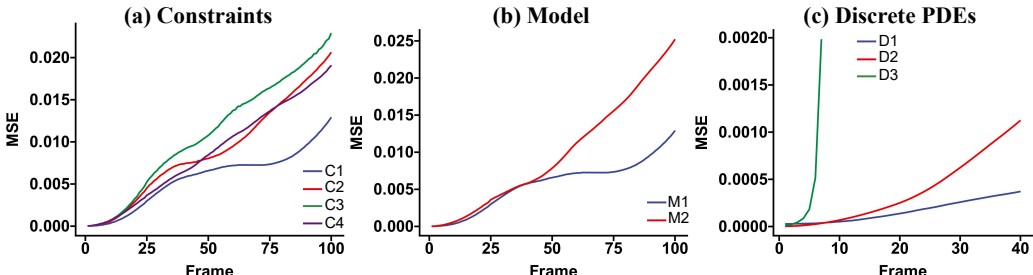

Figure 9: Ablation experiments: (a) constraint, (b) model, and (c) discrete PDEs.

**Constraint Ablation**. In our method, We introduced multiple constraints. We performed experiments by removing the physical and temporal constraints to evaluate the impact on performance. Besides, to validate whether the inclusion of pressure $p$ is beneficial for prediction, we also designed an experiment: no Velocity-Pressure Constraint, which removes $e_1$ in Eq. 12. Results in Figure 9 (a) and Table 3 show that the inclusion of constraints and $p$ is beneficial for prediction.

**Model Ablation**. We introduced a correction network to refine the gradient operator. We conducted experiments to evaluate performance without this network. The results are shown in Figure 9 (b). The correction network notably improves accuracy, especially in later predicted frames.

**Discrete PDEs Ablation**. In Eq. 4, the prediction operator use $c(t')$ to predict, which might be replaced with $c(t_k)$. We conducted two sets of experiments: one where $c(t')$ replace $c(t_k)$ and another involving modifications to the prediction operator. The results, presented in Figure 9 (c), show that while these adjustments improve accuracy in the initial frames, the error increases significantly as the number of predicted frames increases.

Figure 10 illustrates our model's performance during testing alongside its memory consumption. Our model employs a 3D structure in both 2D and 3D scenarios, this results in higher memory usage in 2D scenes. Conversely, in 3D scenes, our model exhibits relatively lower memory usage.

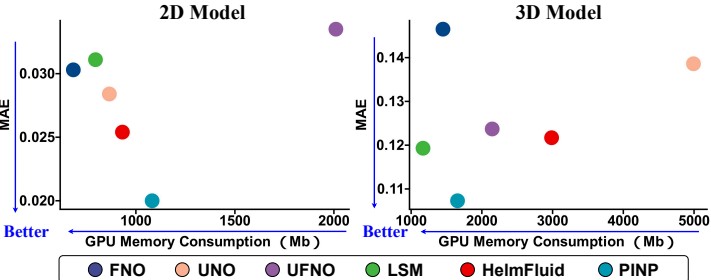

Figure 10: Comparison of GPU memory consumption and MAE

## 6 CONCLUSION

This paper introduces the Physics-Informed Neural Predictor (PINP) model, designed for predicting spatiotemporal evolution. The PINP model leverages latent multi-quantity modeling and integrates PDEs directly into the neural network framework, enabling enhanced understanding and accurate prediction of dynamical systems.

A key feature of the PINP model is its ability to estimate latent physical quantities that are not directly observed in the data. This provides more comprehensive analysis of the physical processes underlying the observed dynamics. The model's architecture embeds the governing PDEs, ensuring that physical constraints are adhered to during both training and prediction phases. This integration leads to improved consistency with physics, facilitating robust long-term forecasts.

When evaluated against a range of baseline methods, the PINP demonstrates better prediction accuracy, particularly in its ability to extrapolate over extended temporal horizons and generalize spatially. This was confirmed through testing on multiple benchmark datasets, where PINP's predictions aligned more closely with ground truth than those of competing approaches. In future research, we aim to expand the application of the PINP model to more complex and comprehensive tasks, accounting for other type of governing PDEs.

ACKNOWLEDGMENTS

The work is supported by the National Natural Science Foundation of China (No. 92270118 and No. 62276269), the Beijing Natural Science Foundation (No. 1232009), and the Strategic Priority Research Program of the Chinese Academy of Sciences (No. XDB0620103). In addition, H.S and Y.L. would like to acknowledge the support from the Fundamental Research Funds for the Central Universities (No. 202230265 and No. E2EG2202X2). Our source codes are available in the following GitHub repository: https://github.com/intell-sci-comput/PINP.

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

APPENDIX

# A   IMPLEMENTATION DETAILS

In this section, we provide additional details on key aspects of the paper, including the Gradient Operator, Signed Distance Field (SDF), Mask Settings and Model Structure.

For the gradient operator, we discuss its approximation and the associated errors, particularly in fluid-boundary interactions. To address these issues, we introduce specific masks—Mask1 for the first-order gradient operator and Mask2 for the second-order operator—to reduce these errors.

The Signed Distance Field (SDF) helps define boundary points and enhances spatial understanding, which is crucial for handling irregular shapes and obstacles. The SDF also plays a key role in the boundary point identification process (Figure S1).

Considering that the established mapping function in Eq. 8 is nonlinear and multiscale, our architecture employs a structure similar to 3D U-Net to build complex mappings and multiscale features.

These detailed approaches contribute to improving the accuracy and robustness of the overall prediction model.

## A.1   GRADIENT OPERATOR

Using the Navier-Stokes equations, we transform the partial derivative of concentration with respect to time into spatial gradient terms. For these gradient terms, we employ the second-order central difference method to achieve estimation. Letting $x$ be an interior point with $x - \Delta x$ and $x + \Delta x$ be points neighboring it to the left and right respectively:

$$
\begin{aligned}
f\left(x + \Delta x\right) &= f(x) + \Delta x f'(x) + \Delta x^2 \frac{f''(x)}{2} + \Delta x^3 \frac{f'''(\xi_1)}{6}, \xi_1 \in (x, x + \Delta x) \\
f\left(x - \Delta x\right) &= f(x) - \Delta x f'(x) + \Delta x^2 \frac{f''(x)}{2} - \Delta x^3 \frac{f'''(\xi_2)}{6}, \xi_2 \in (x, x - \Delta x).
\end{aligned} \tag{S1}
$$

Solving the linear system, we derive:

$$
f'(x) \approx \frac{f\left(x + \Delta x\right) - f\left(x - \Delta x\right)}{2\Delta x}. \tag{S2}
$$

To achieve better optimization results, we exclude gradients of boundary points from the training loss calculation.

## A.2   SIGNED DISTANCE FUNCTION

A signed distance field (SDF) measures the orthogonal distance from a given point $x$ in a metric space to the boundary of a set $\Omega$, with the sign determined by the location of $x$ relative to $\Omega$. Specifically: If $x$ is inside $\Omega$, the signed distance is positive. The distance decreases as $x$ approaches the boundary of $\Omega$. If $x$ is outside $\Omega$, the signed distance becomes negative. In essence, the signed distance field provides both a magnitude of the shortest distance to the boundary of the set and a sign indicating whether the point is inside or outside the boundary.

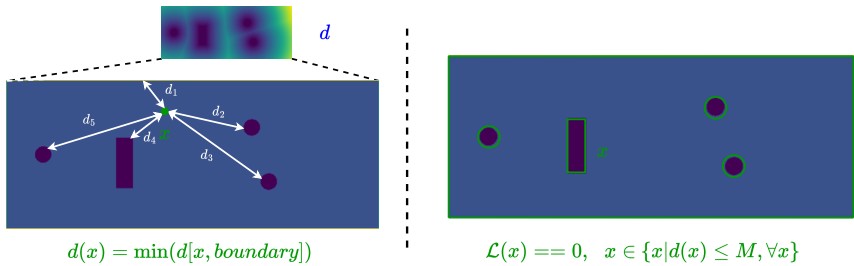

$$d(x) = \min(d[x, boundary]) \qquad\qquad \mathcal{L}(x) == 0, \quad x \in \{x | d(x) \le M, \forall x\}$$

Figure S1: An example of Signed distance function and Mask

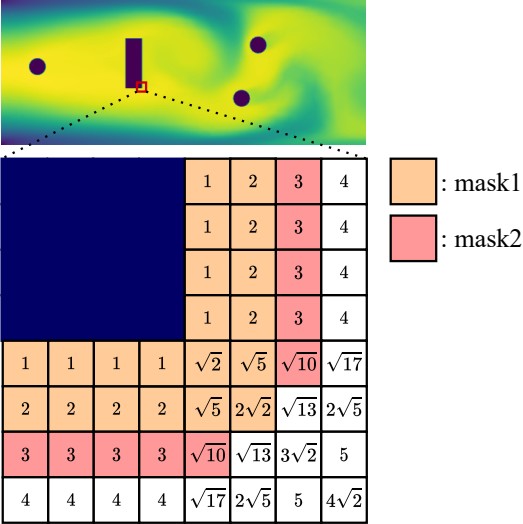

Figure S2: Mask1 and Mask2.

In this paper, we define $\Omega$ as the set of external boundaries and obstacle boundaries. Since no fluid flows through the obstacles, we do not assign negative values to the points inside the obstacles; instead, we set them to zero.

In addition to providing richer spatial information, the directed distance field helps us identify boundary points. As mentioned earlier, to achieve better optimization results, we exclude the gradients of boundary points from the training loss calculation. While boundary points on the external boundary are easy to identify, determining the boundary points for internal obstacles, especially those with irregular shapes, is more challenging. This problem can be addressed using the directed distance field. We define the orthogonal distance from point $x$ to the boundary of set $\Omega$ as $d$. Points with a distance less than $M$ from the boundary are considered boundary points. Therefore, the set of boundary points is given by $\{x|d(x) \le M, \forall x\}$.

Figure S1 illustrates an example, including the definition of distance(left) and the setting of the loss at boundary points to zero(right).

## A.3 MASK SETTINGS

The specific boundary handling is illustrated in Figure S2. Since the gradient operator in Formula S2 introduces approximations, these approximation errors become more pronounced during fluid-boundary interactions. To reduce such errors, we employ a masking approach. For first-order gradient operators, we define Mask1:

$$\text{Mask1} := \{x|d(x) \le 2.5, \forall x\}. \tag{S3}$$

and for second-order gradient operators, we define Mask2:

$$\text{Mask2} := \{x|d(x) \le 3.5, \forall x\}. \tag{S4}$$

This helps ensure better handling of boundary interactions and minimizes the impact of approximation errors in fluid simulations.

## A.4 MODEL STRUCTURE

The model adopts a recurrent prediction structure, utilizing a sliding window approach for forecasting. Each time, it predicts one frame based on the previous four frames and incorporates the predicted frame into the window for subsequent predictions. The prediction architecture for each step is illustrated in Figure 2.

In Figure 2, we use Physical Inference Neural Network to establish the mapping function from past data to the physical quantities at time $t'$. Considering that the established mapping function is

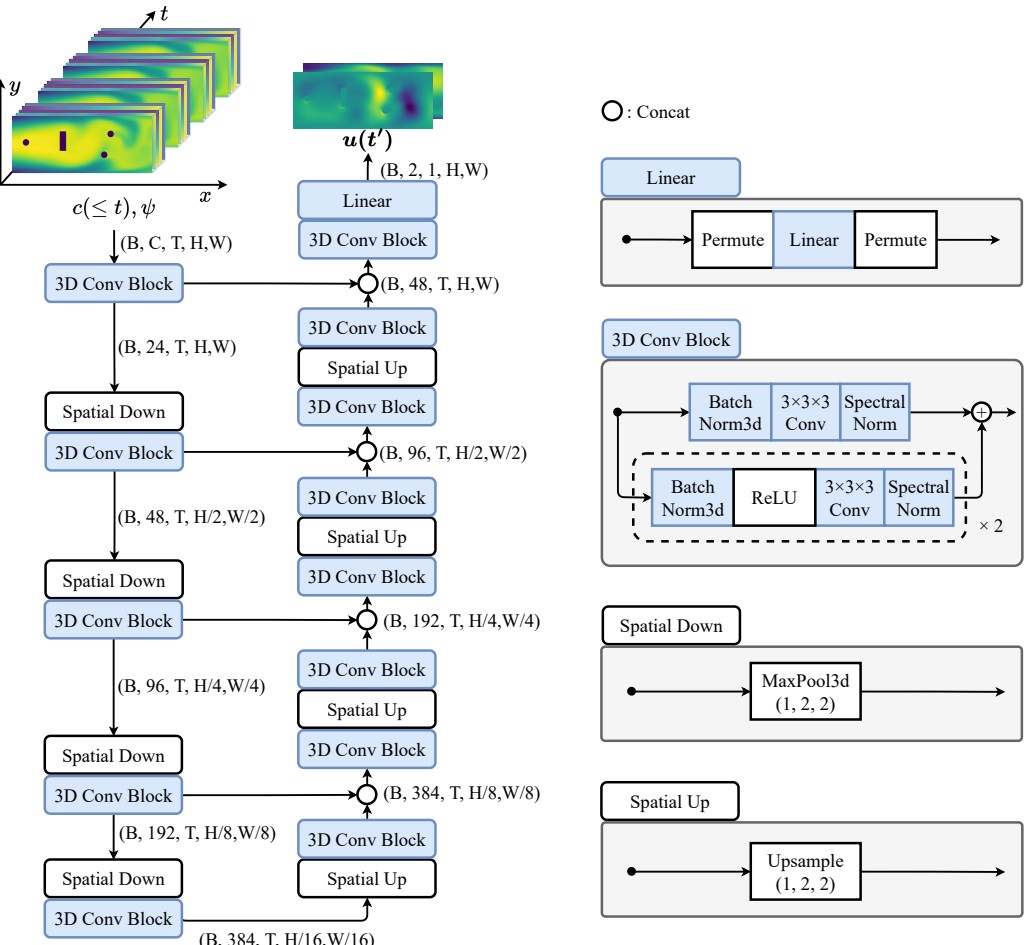

Figure S3: Detailed structure of our model, using the complex mapping from $c(\leq t), \psi$ to $u(t')$ as an example.

nonlinear and multiscale, our architecture employs a structure similar to 3D U-Net to build complex mappings and multiscale features.

We present the detailed structure of our model, using the complex mapping from $c(\leq t), \psi$ to $u(t')$ as an example (Figure S3). The structures of the remaining mapping functions are similar to this.

## A.5 NOWCASTNET SETTING AND RESULT

It is important to note that NowcastNet utilizes a deterministic prediction model (Evolution network) combined with a probabilistic generative model (GAN). Given that this is a deterministic prediction task, the Evolution network outperforms the combined generative model (GAN) in most evaluation metrics. In the main text, we selected the best prediction results from the deterministic model. In this section, we provide the complete results.

## B DATA AND EVALUATION

In this section, we provide a detailed explanation of the data and evaluation metrics, including details such as the data generation method, data scale, and the specifics of the prediction tasks.

Table S1: Comparison of NowCastNet results under different settings with Our Model

| Method | MSE (1e-3) | CSI-M | CSI-219 | CSI-181 | CSI-160 | CSI-133 | CSI-74 | CSI-16 |
|---|---|---|---|---|---|---|---|---|
| pySTEPS (2019) | 8.4682 | 0.3373 | 0.0912 | 0.1756 | 0.2162 | 0.3244 | 0.5848 | 0.6313 |
| Evolution network (wrap mode = 'bilinear') | **3.8883** | 0.3743 | 0.0803 | 0.1811 | 0.2351 | 0.3515 | **0.6635** | **0.7339** |
| Evolution network (wrap mode = 'nearest') | 6.1783 | 0.3529 | 0.0971 | 0.1858 | 0.2220 | 0.3110 | 0.5986 | 0.7027 |
| NowcastNet (2023) | 5.7999 | 0.3549 | 0.0933 | 0.1844 | 0.2179 | 0.3104 | 0.6131 | 0.7106 |
| PINP (this work) | 4.1684 | **0.3800** | **0.0989** | **0.1952** | **0.2448** | **0.3573** | 0.6556 | 0.7279 |

Table S2: Data description includes spatial resolution, number of training samples, and training/testing setup:

| Data | Spatial Resolution | Training datasets | Training | Testing |
|---|---|---|---|---|
| Flow 2D | $640 \times 256$ | 2500 | 4 to 4 | 4 to 40 |
| Smoke 2D | $256 \times 256$ | 1500 | 4 to 10 | 4 to 100 |
| Smoke 3D | $64 \times 64 \times 64$ | 340 | 4 to 4 | 4 to 10 |
| SEVIR | $96 \times 96$ (down) | 35718 | 13 to 12 | 13 to 12 |

## B.1 DATASET

**Fluid 2D**. In this study, the dataset used consists of image sequences of fluid-driven mass diffusion, generated using the Computational Fluid Dynamics (CFD) method through COMSOL software (Multiphysics, 1998). The physical setup involves an active flow field of width $W$ and height $H$, filled with a fluid of density $\rho$ and viscosity coefficient $\mu$. At the left side of the field is the input, where a material with a diffusion coefficient $D_c$ and concentration $c$ flows and diffuses, driven by an initial velocity $u$. The temperature is set to standard conditions, and the pressure is one atmosphere. The data used in the experiments consists only of concentration data. The parameters for data generation are shown in Figure S4.

This dataset, generated by COMSOL, is used for training and evaluating the model. The training dataset consists of five subsets, each containing 500 concentration images representing the diffusion of materials in the fluid. These images form a time series, with each image having a resolution of $640 \times 256$ pixels. The test dataset is divided into seven subsets, each containing 200 images of the same resolution, also forming time series data. Differences in physical field parameters across the subsets are primarily reflected in changes in obstacle positions.

To evaluate the spatial generalization ability of the model, the test set includes three special subsets: one involves changes in the shape of obstacles, another simulates the removal of an obstacle, and the third adds a new obstacle to the original physical field. These conditions are absent in the training data to assess the model's generalization performance and to determine if it has successfully learned the underlying physics.

To evaluate the model's temporal extrapolation ability, we used a strategy during training where the model predicts the subsequent four frames based on the first four frames. In the testing phase, the model is tasked with predicting up to 40 frames using the first four frames.

**Smoke 2D**. Compared to the fluid dataset, the smoke dataset features more complex scenes. We focus on the process of smoke rising and diffusing in the air, with a square boundary defining the space. The data was generated using $\Phi_{\text{Flow}}$ (Holl & Thuerey, 2024), with each image having a resolution of $256 \times 256$ pixels, and the location of the smoke source varies between the training and test sets. The data used in the experiments consists only of concentration data.

Similar to the fluid dataset, we use a strategy where the model predicts the next ten frames based on the first four frames during the training phase. In the testing phase, the model is required to predict up to 100 frames using the initial four frames. Different data has varying smoke source location.

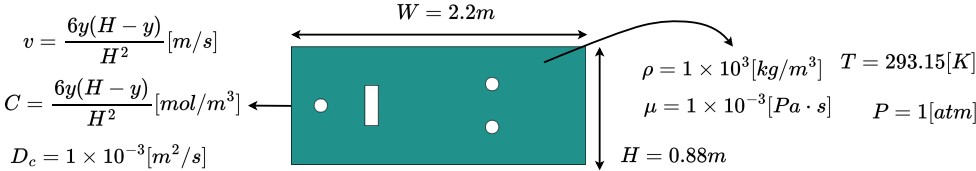

Figure S4: The configuration chart for data generation settings.

**Smoke 3D**. We extend the 2D smoke dataset to a 3D setting, considering the process of smoke ascending and expanding in a confined 3D space with a resolution of $64 \times 64 \times 64$. The boundary is defined as a cube, and the location of the smoke source varies between the training and test sets. The data used in the experiments consists only of concentration data.

Similar to the fluid dataset, we use a strategy where the model predicts the next four frames based on the initial four frames during the training phase. In the testing phase, the model is required to predict up to 10 frames using the first four frames. We used FactFormer (Li et al., 2023a) to generate 340 sets of data for training. Different data has varying smoke source location.

**SEVIR**. Storm Event Imagery (SEVIR) (Veillette et al., 2020) is a dataset encompassing thousands of meteorological events, including various storms, lightning strikes, and precipitation events in the United States from 2017 to 2019. Each event is documented with image sequences covering a 4-hour period over a 384 km × 384 km area. The dataset includes data from five sensors: satellite imagery, infrared (water vapor), infrared (window), NEXRAD radar composites of Vertically Integrated Liquid (VIL), and lightning data. All sensors provide data with a spatial coverage of 384 km × 384 km and a temporal resolution of 5 minutes.

For nowcasting, we selected the NEXRAD radar VIL composites. The VIL data has a spatial resolution of 1 km and is recorded at 5-minute intervals. Following the approach of Gao et al. (2022b), we use 65 minutes of VIL data (13 frames) to predict up to 60 minutes ahead (12 frames) for precipitation nowcasting. Due to computational limitations, we downsample the spatial resolution to $96 \times 96$. This downsampled dataset includes 35,718 training samples, 9,060 validation samples, and 12,159 test samples.

The data overview is presented in Table S2, including spatial resolution, number of training samples, and training and testing setup.

### B.2 EVALUATION METRICS

In the simulated data evaluation, MSE and MAE are used as the evaluation indexes. In the real data evaluation, for the adjacent precipitation prediction index, we follow Veillette et al. (2020), and use the Critical Success Index (CSI) to evaluate the prediction quality.

Given predictions $\{\widehat{x}_{n,t,h,w}\}$ and corresponding ground truth $\{x_{n,t,h,w}\}$, $\widehat{x}_{n,t,h,w}, x_{n,t,h,w} \in \mathbb{R}$, $n = 1, \ldots, N$, $t = 1, \ldots, T$, $h = 1, \ldots, H$, $w = 1, \ldots, W$, the above-mentioned metrics can be calculated as follows:

$$\text{MSE} = \frac{1}{NTHW} \sum_{n=1}^{N} \sum_{t=1}^{T} \sum_{h=1}^{H} \sum_{w=1}^{W} \|x_{n,t,h,w} - \widehat{x}_{n,t,h,w}\|_2^2$$

$$\text{MAE} = \frac{1}{NTHW} \sum_{n=1}^{N} \sum_{t=1}^{T} \sum_{h=1}^{H} \sum_{w=1}^{W} \|x_{n,t,h,w} - \widehat{x}_{n,t,h,w}\|_1 .$$

(S5)

For the inferred and predicted latent physical quantities, referring to Kochkov et al. (2021), we quantitatively evaluate them by calculating the Correlation.

Table S3: Hyperparameter and Training setup

| Type | Name | Meaning | Value |
|------|------|---------|-------|
| Hyperparameter | Mask1 | First-order gradient operator mask | Eq. S3 |
| Hyperparameter | Mask2 | Second-order gradient operator mask | Eq. S4 |
| Training setup | LR | Learning Rate | 1e-3 |
| Training setup | Epoch | Number of training epochs | 100 |
| Training setup | Optimizer | Type of model optimizer | Adam |
| Training setup | Scheduler | Schedule the learning rate of the optimizer | StepLR |
| Training setup | Batch Size | the number of samples processed together | $(2, 2, 2, 32)^*$ |

$()^*$: Value is different for each dataset, in the order of 2D fluid, 2D smoke, 3D smoke, and SEVIR.

$$\text{Correlation} = \frac{\sum_{i=1}^{n} \left( \widehat{x}_{n,t,h,w} - \overline{\widehat{x}_{n,t,h,w}} \right) (x_{n,t,h,w} - \overline{x_{n,t,h,w}})}{\sqrt{\sum_{i=1}^{n} \left( \widehat{x}_{n,t,h,w} - \overline{\widehat{x}_{n,t,h,w}} \right)^2} \sqrt{\sum_{i=1}^{n} \left( x_{n,t,h,w} - \overline{x_{n,t,h,w}} \right)^2}} \tag{S6}$$

Specifically, Veillette et al. (2020) used six precipitation thresholds which correspond to pixel values $[219, 181, 160, 133, 74, 16]$. The prediction $\widehat{x}_{n,t,h,w}$ and the ground-truth $x_{n,t,h,w}$ are rescaled back to the range $0 - 255$.

$$\#\text{Hits}(\tau) = \#\{\widehat{x}_{n,t,h,w} \mid \widehat{x}_{n,t,h,w} \geq \tau, \ x_{n,t,h,w} \geq \tau\}$$
$$\#\text{Misses}(\tau) = \#\{\widehat{x}_{n,t,h,w} \mid \widehat{x}_{n,t,h,w} \geq \tau, \ x_{n,t,h,w} < \tau\} \tag{S7}$$
$$\#\text{F.Alarms}(\tau) = \#\{\widehat{x}_{n,t,h,w} \mid \widehat{x}_{n,t,h,w} < \tau, \ x_{n,t,h,w} \geq \tau\},$$

$$\text{CSI} - \tau = \frac{\#\text{Hits}(\tau)}{\#\text{Hits}(\tau) + \#\text{Misses}(\tau) + \#\text{F.Alarms}(\tau)}$$
$$\text{CSI-M} = \frac{1}{6} \sum_{\tau} \text{CSI-} \tau, \ \tau \in [219, 181, 160, 133, 74, 16]. \tag{S8}$$

$\#$ represents the number of elements in the set, $\tau \in [219, 181, 160, 133, 74, 16]$ is one of the thresholds. We denote the average CSI-$\tau$ over the thresholds $[219, 181, 160, 133, 74, 16]$ as CSI-M.

It must be stated that the extreme point of MSE and the extreme point of CSI are usually not consistent, and the model with the smallest MSE on the validation set is selected for testing.

## C  TRAINING DETIALS

In this section, we present the training details of our experiments, including hyperparameter settings, training setup, and the specifics of multi-loss training.

### C.1  HYPERPARAMETERS AND TRAINING SETUP

The hyperparameters and experimental setup used in our experiments are shown in the table below:

### C.2  MULTI-LOSS FUNCTION TRAINING

Since the model training process involves a multi-loss optimization problem (Eq. 16), we further show the training details.

Figure S5 (left) shows the change of Loss with the number of training rounds during training, and Figure S5 (right) shows the change of validation metrics with the number of training rounds.

As shown in Figure S5 (left) and Figure S5 (right) , the loss corresponding to the data constraint remains the largest, indicating that despite the introduction of multiple loss functions, the other losses do not significantly affect the primary constraint. Additionally, the loss associated with the temporal

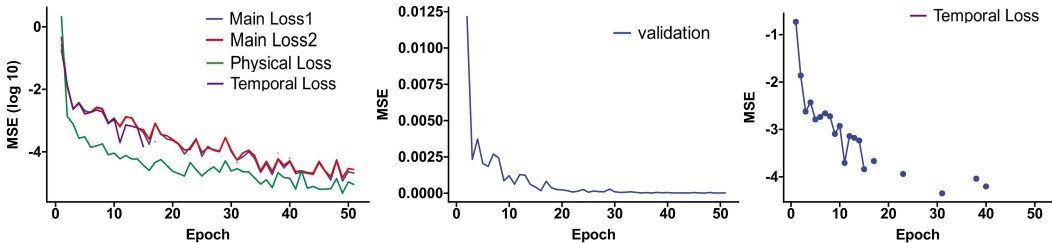

Figure S5: Training loss versus training epochs (left), loss of validation versus training epochs (middle) and Temporal loss versus training epochs (right).

Table S4: Correlation between Predicted and True Latent Physical Quantities.

| Latent Physical Quantities | Correlation (PINP) | Correlation (HelmFluid) |
|---|---|---|
| $u_x$ | 0.9017 | 0.8891 |
| $u_y$ | 0.9109 | 0.8997 |
| $p$ | 0.8984 | - |

constraint reaches a value of zero, demonstrating that we can effectively bound the temporal loss in $||c(\boldsymbol{x}, t+1) - c(\boldsymbol{x}, t)||_2^2$, preventing it from degrading into:

$$\mathcal{L}_{\text{Temporal}}(\boldsymbol{x}, t) = ||c(\boldsymbol{x}, t') - c(\boldsymbol{x}, t)||_2^2. \tag{S9}$$

As shown in Figure S5 (middle), the test metrics of validation decrease rapidly with the increase in training epochs and then level off. In a sense, the physical constraints and temporal constraints act as a form of regularization, providing some resistance to overfitting.

## D    DISCUSSION ON LATENT PHYSICAL QUANTITIES

In this section, we will compare our estimated latent physical quantities with the baselines and provide a detailed explanation.

Since the HelmFluid model is capable of estimating the velocity field (but not the pressure field), we considered it a relevant comparison for our method. The comparative results of correlation on the estimated latent physical quantities are shown in Table S4. We can see that our estimation subpasses that of HelmFluid.

Additionally, it's important to note that due to the unknown physical scale (since only the grayscale video sequences of the concentration field are provided as the training data), our estimated velocity fields and the true values have a scaling relationship, expressed as follows:

$$\hat{\boldsymbol{u}} = \alpha\boldsymbol{u}, \tag{S10}$$

where $\alpha$ is a scaling constant related to the actual physical scale. When only concentration videos are available, we cannot determine the exact physical scale of the region represented in the video (e.g., the scale may lie in any range such as from 10 cm to 1 m in the smoke dataset). This uncertainty introduces a constant factor difference between the predicted and true values.

Thus, directly computing the MSE might be inappropriate. To address this, we considered two methods for calibration:

- Case 1: Using the true initial velocity values (assumed known a priori) to compute $\alpha$.
- Case 2: Using just one true data point to estimate $\alpha$.

We then apply the computed $\alpha$ to correct the estimated results (note that HelmFluid also requires calibration). The calibrated MSE values are shown in Table S5.

Table S5: Comparison of calibrated MSE for latent physical quantities between our model and HelmFluid.

| Latent Physical Quantities | PINP (Case 1) | PINP (Case 2) | HelmFluid (Case 1) | HelmFluid (Case 2) |
|---|---|---|---|---|
| $u_x$ | 0.1056 | 0.3128 | 0.1482 | 0.5035 |
| $u_y$ | 0.0806 | 0.2806 | 0.1187 | 0.3407 |
| $p$ | 0.1539 | 0.4653 | - | - |

Table S6: Comparison of our model's results with those obtained after removing the physical constraints and temporal constraints on 2D fluid data.

| Metrics | no Physical Constraints | no Temporal Constraints | no Velocity-Pressure Constrain | Normal |
|---|---|---|---|---|
| MAE | 0.0209 | 0.0272 | 0.0293 | 0.0262 |
| MSE | 0.0057 | 0.0089 | 0.0108 | 0.0087 |

The results in Tables S4 and S5 highlight the superiority of our approach in predicting the latent physical quantities. Moreover, even with a constant factor difference, a high correlation ($\geq 0.9$) is sufficient to demonstrate the consistency between our predictions and the true data in terms of distribution, regardless of scaling. This can serve as supplementary information for concentration prediction and provide interpretable evidence for the future evolution of fluid dynamics.

## E  ABLATION STUDY

In this section, we conduct ablation studies to assess the contributions of different components of our method, including constraint ablation, model ablation, and prediction operator ablation.

### E.1  CONSTRAINTS

The constraint ablation experiment consists of two parts: removing the physical constraints and removing the temporal constraints:

**Physical Constraints.** We removed the physical constraints while retaining the temporal constraints and retrained the model. The results show a decline in prediction accuracy due to the absence of physical constraints (Table S6), and the inferred latent physical quantities exhibit issues, lacking real physical significance (Figure S6).

**Velocity-Pressure Constraint**. To validate whether the inclusion of pressure $p$ is beneficial for prediction, we also designed an experiment: no Velocity-Pressure Constraint, which removes $e_1$ in Eq. 12. Results in Figure 9 (a) and Table 3 show that the inclusion of constraints and $p$ is beneficial for prediction. In Figure S6, we present this result, where it can be seen that the predicted velocity field is not correct.

**Temporal Constraints.** We removed the temporal constraints while retaining the physical constraints and retrained the model. The results indicate that the prediction accuracy also declined due to the lack of temporal constraints (Table S6), and the inference of latent physical quantities, especially the pressure field, was problematic (Figure S6).

### E.2  MODELS

In this section, we perform ablation experiments on our model structure by removing the correction network to verify its necessity.

**Correction Network.**   As shown in Table S7, after removing the correction network, prediction accuracy significantly decreased. Figure S7 illustrates an example where "stripes" appeared in the output after the correction network was removed. These artifacts are caused by errors in the gradient operator. By using the correction network, this issue can be effectively mitigated.

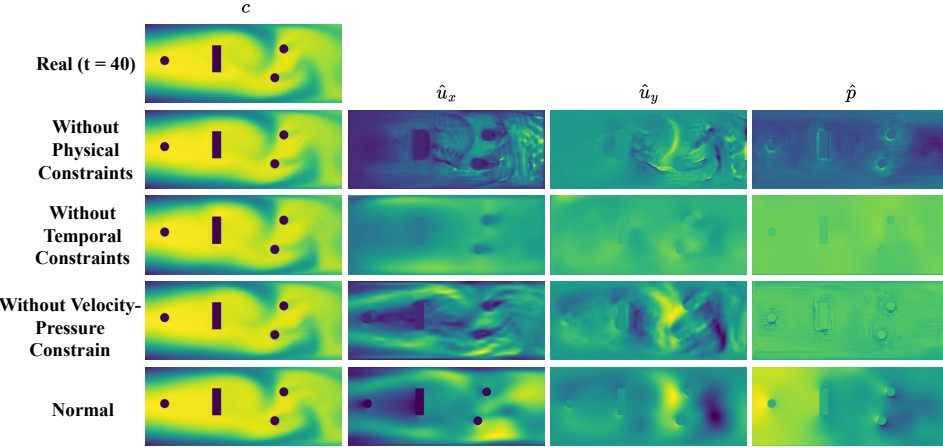

Figure S6: An example comparing the results of our model with those obtained after removing the physical constraints and temporal constraints.

Table S7: Comparison of our model's results with those obtained after removing the Correction Network on 2D smoke data.

| Metrics | Without Correction Network | Normal |
|---------|---------------------------|--------|
| MAE | 0.0316 | 0.0209 |
| MSE | 0.0098 | 0.0057 |

### E.3  DISCRETE PDEs

In this section, we conduct ablation experiments on the discrete PDE operators, including modifying the discrete PDE operators and replacing $c(t')$ with $c(t)$.

Table S8: Comparison of our model's results with those obtained after changing Discrete PDEs and replacing $c(t')$ with $c(t)$ on 2D smoke data.

| Metrics | Change Discrete PDEs | Replacing $c(t')$ with $c(t)$ | Normal |
|---------|---------------------|-------------------------------|--------|
| MAE | 0.0197 | Inf | 0.0107 |
| MSE | 0.0010 | Inf | 0.0003 |

**Discrete PDEs.** We modified the discrete PDE operators, with the original operator:

$$\hat{c}'(\boldsymbol{x}, t+1) = c(\boldsymbol{x}, t) + \left(-\boldsymbol{u}(\boldsymbol{x}, t') \cdot \nabla c(\boldsymbol{x}, t') + \mathrm{Pe}^{-1}\nabla^2 c(\boldsymbol{x}, t')\right)\Delta t, \tag{S11}$$

and the modified version:

$$\hat{c}'(\boldsymbol{x}, t+1) = c(\boldsymbol{x}, t) + u_x(\boldsymbol{x}, t') + u_y(\boldsymbol{x}, t'). \tag{S12}$$

The comparison of results before and after the modification is shown in Table S8, where the prediction accuracy significantly decreased after the modification.

**Replacing $c(t')$ with $c(t)$.** In the prediction operator, we use the physical quantities at time $t'$ along with the discrete PDE prediction operator for forecasting, constraining the observable data between $t$ and $t+1$. If we directly use the observable data at time $t$ to guide the prediction, the experimental results, as shown in Table S8, indicate that errors accumulate rapidly when predicting beyond the training time steps. Figure S8 show an example.

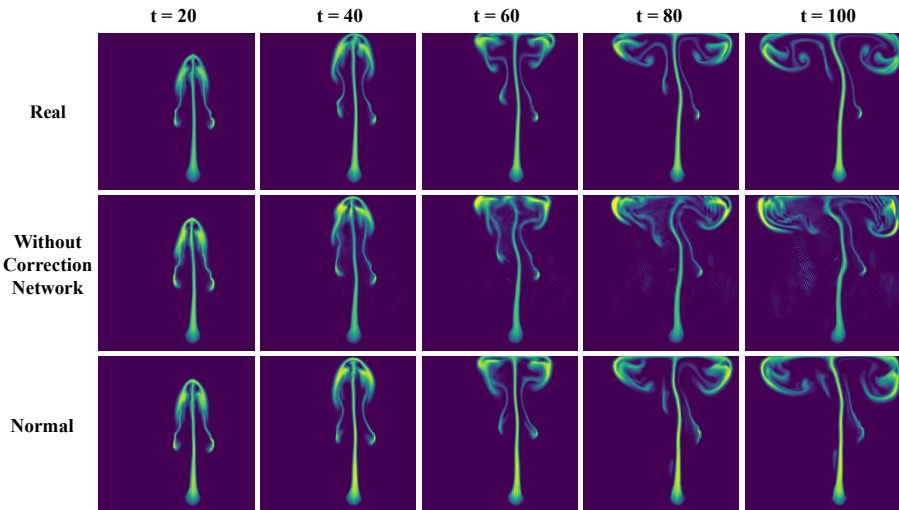

Figure S7: An example comparing the results of our model with those obtained after removing the Correction Network.

Table S9: Comparison of Evaluation Metrics on the SEVIR dataset under Different Physical Loss Configurations

| Method | MSE (1e-3) | CSI-M | CSI-219 | CSI-181 | CSI-160 | CSI-133 | CSI-74 | CSI-16 |
|---|---|---|---|---|---|---|---|---|
| Lowered Physical Loss Weight | 4.0305 | 0.3662 | 0.0781 | 0.1724 | 0.2213 | 0.3402 | 0.6563 | 0.7288 |
| No Physical Loss | 4.0092 | 0.3569 | 0.0674 | 0.1541 | 0.2081 | 0.3274 | 0.6547 | 0.7299 |
| PINP (this work) | 4.1684 | 0.3800 | 0.0989 | 0.1952 | 0.2448 | 0.3573 | 0.6556 | 0.7279 |

### E.4 PHYSICAL LOSS ANALYSIS ON THE SEVIR DATASET.

The nowcasting scenario may not strictly adhere to the NS equations. However, we believe that incorporating the NS equations helps to capture the motion characteristics of nowcasting.

We conducted the following experiments: (1) Lowered the physical loss weight (weight = 0.1) while keeping other settings unchanged. (2)Removed the physical loss entirely.

The results are showned on Table S9, which demonstrate that while nowcasting processes do not strictly adhere to the NS equations, these equations effectively capture certain underlying motion characteristics. When the physical loss weight is reduced, loosening the constraints, the MSE decreases. However, this relaxation comes at the cost of reduced ability to model essential motion features, as reflected in the decline of the CSI.

## F ADDITIONAL RESULTS

In this section, we provide additional details on the experimental results presented in the main text. This includes supplementary results for multi-physical quantity inference (Figure S9, Figure S10), a comparison between our predicted velocity field and the velocity field predicted by NowcastNet Figure S11, a detailed comparison between our method and baselines on the 2D fluid data (Figure S12, Figure S13), 2D smoke data (Figure S14), 3D smoke data (Figure S15, Figure S16), and SEVIR data (Figure S17, Figure S18).

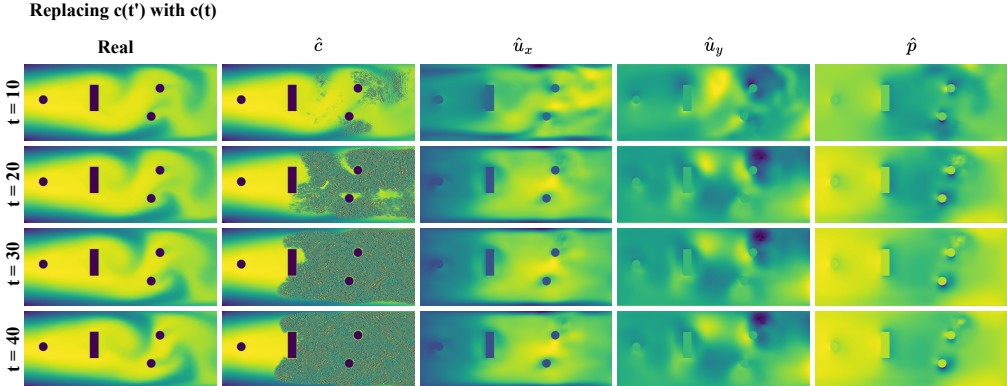

Figure S8: An example showing the results after replacing $c(t')$ with $c(t)$.

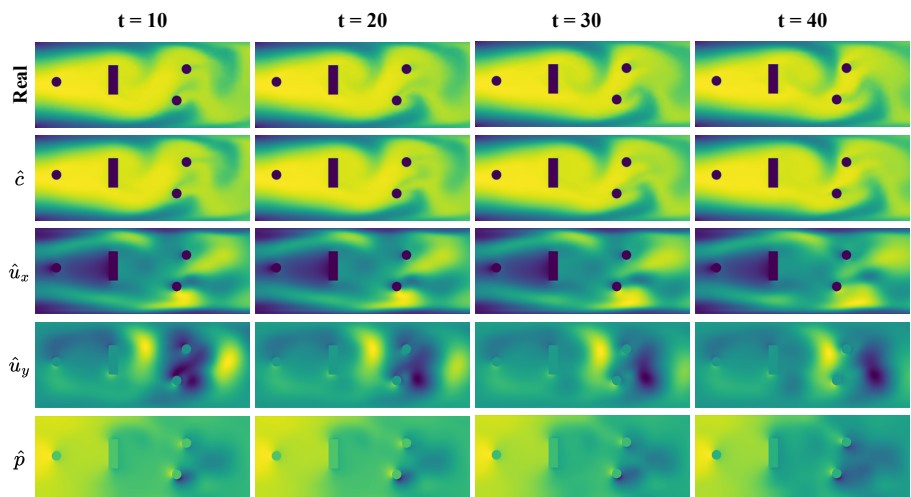

Figure S9: An example of 2D fluid prediction and latent physical quantity inference.

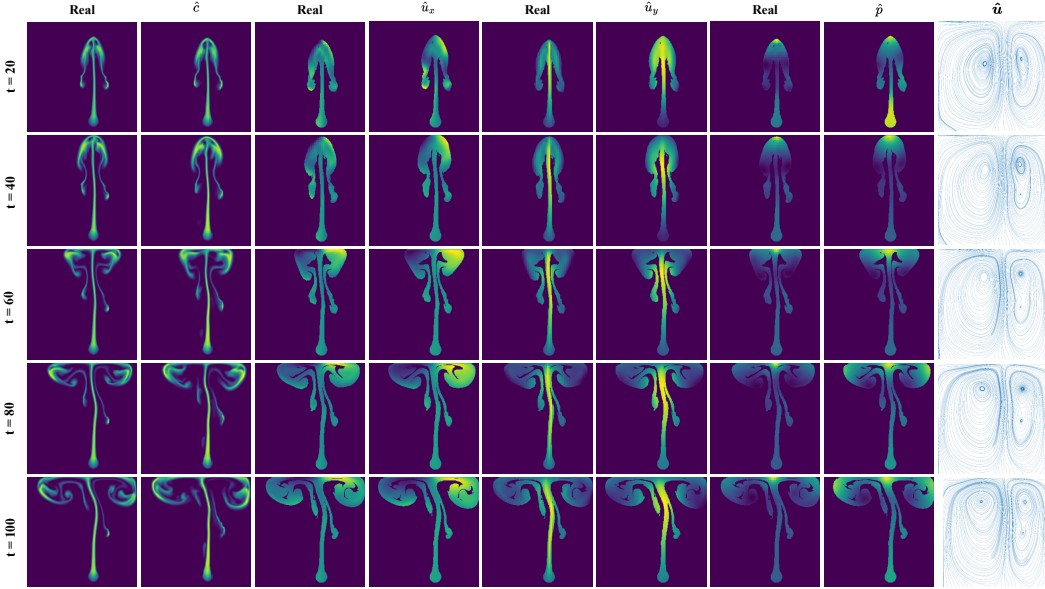

Figure S10: An example of 2D smoke prediction and latent physical quantity inference.

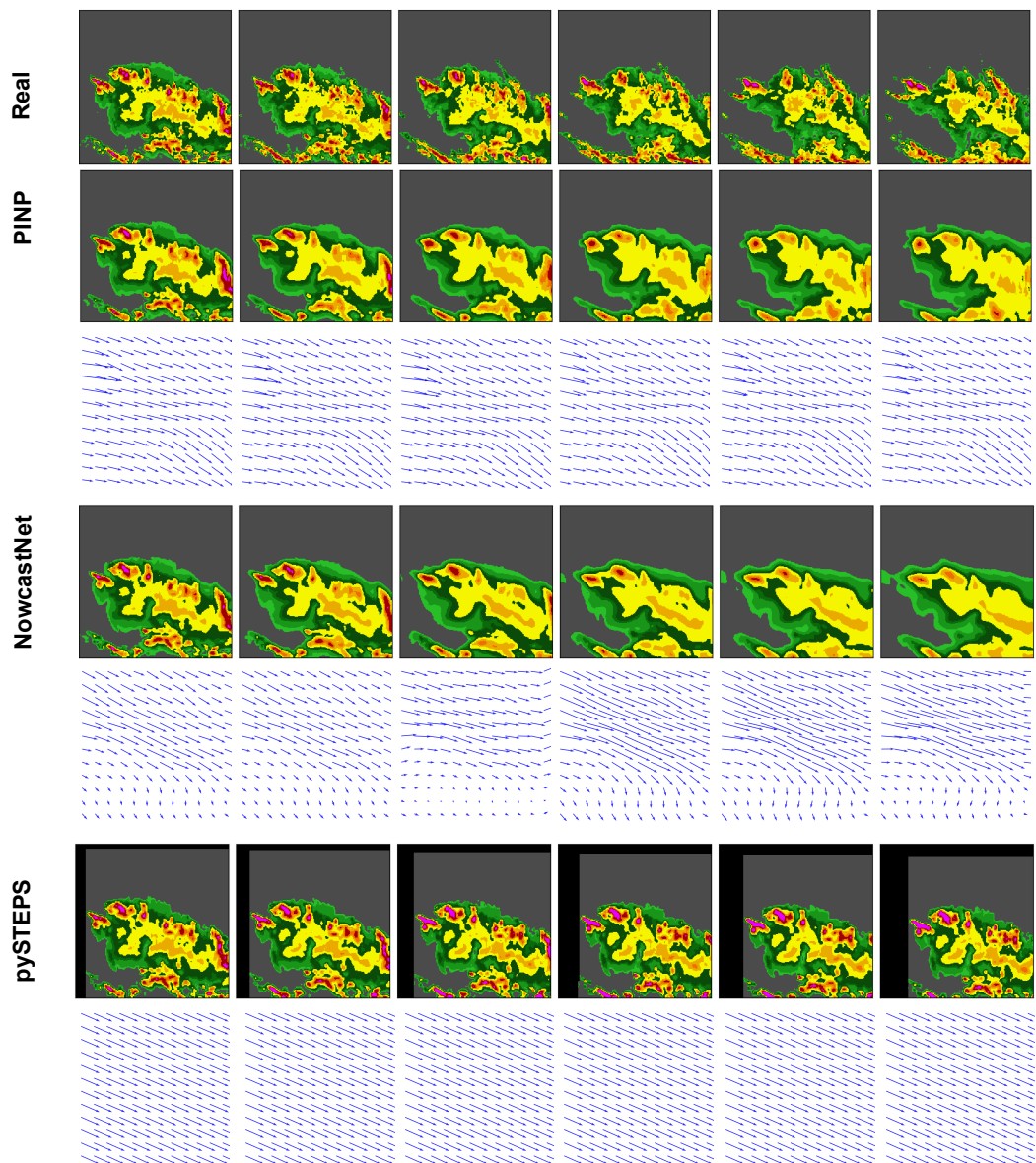

Figure S11: The Comparison Between Our Predicted Velocity Field and the Velocity Field Predicted by NowcastNet and pySTEPS. The black regions in pySTEPS represent NaN values.

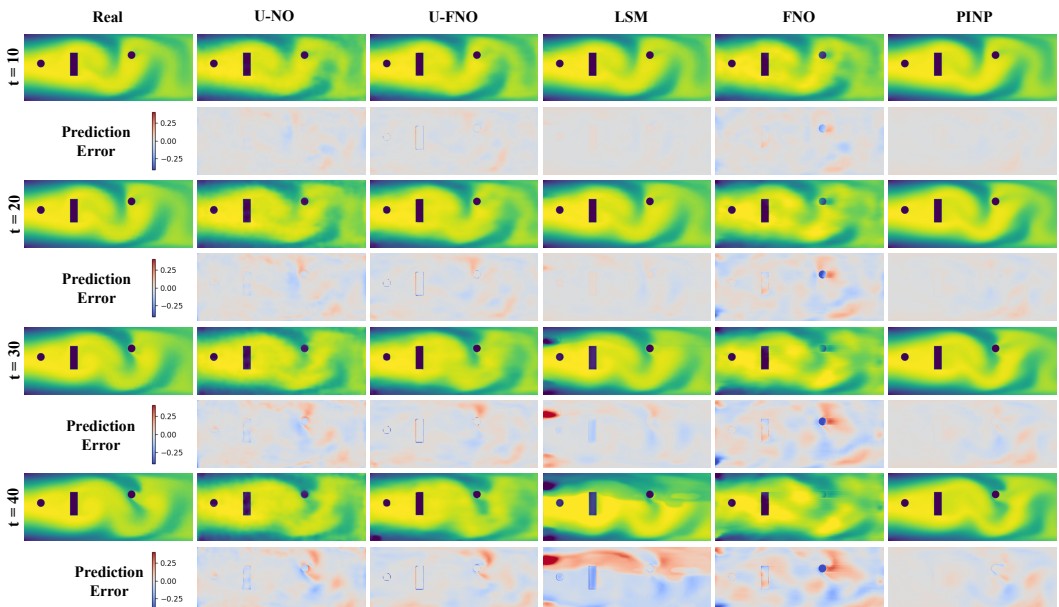

Figure S12: An example comparing our model with the baseline on 2D fluid data.

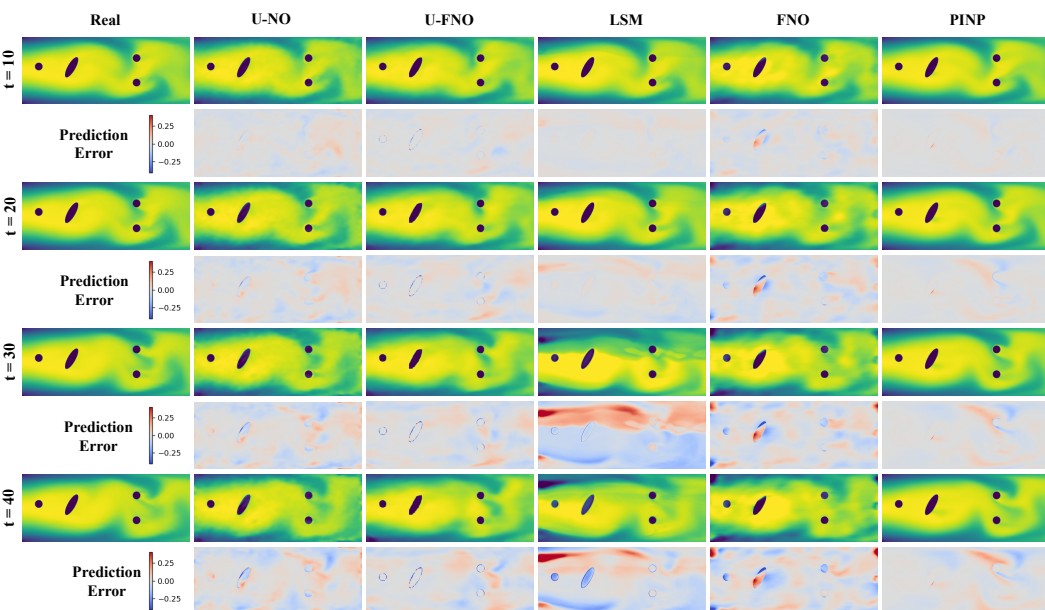

Figure S13: An example comparing our model with the baseline on 2D fluid data.

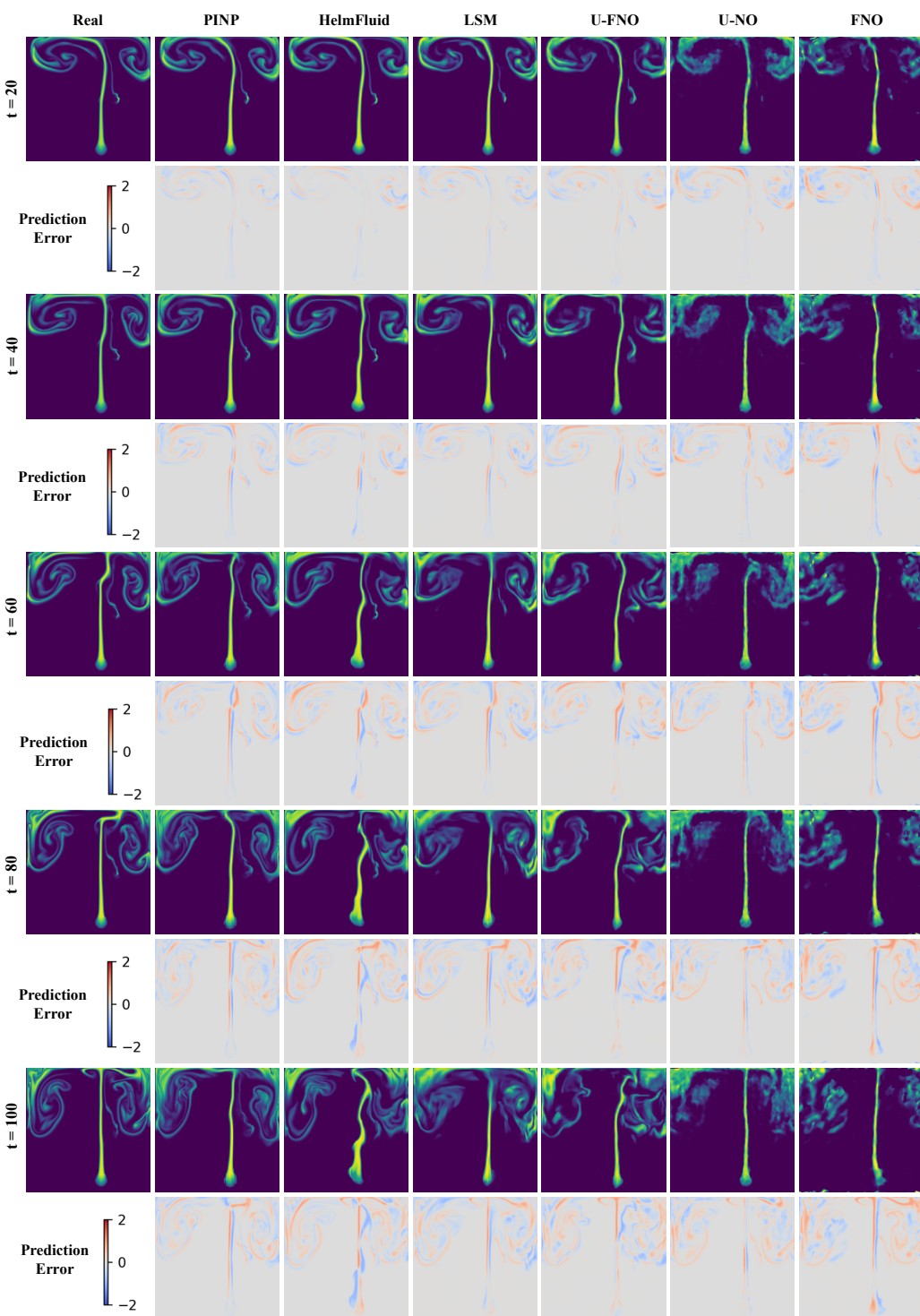

Figure S14: An example comparing our model with the baseline on 2D smoke data.

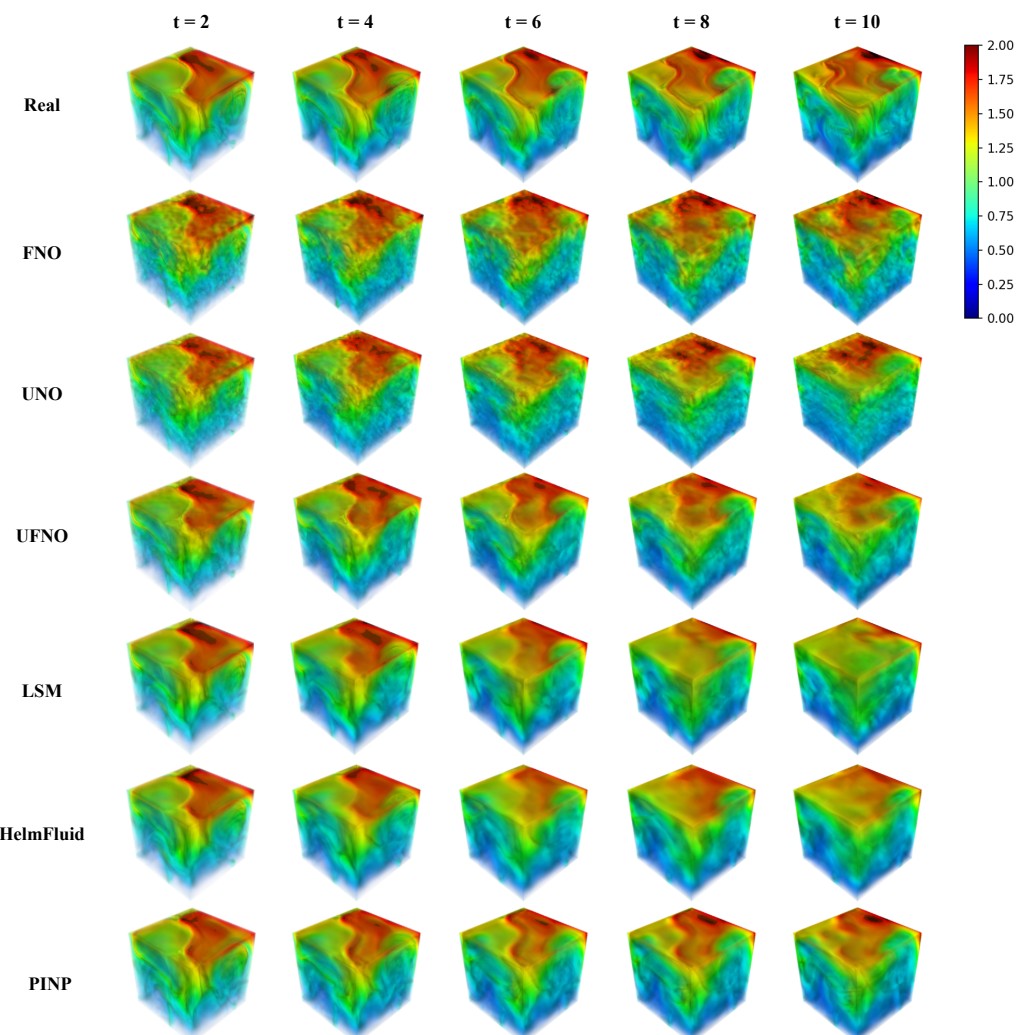

Figure S15: An example comparing our model with the baseline on 3D smoke data.

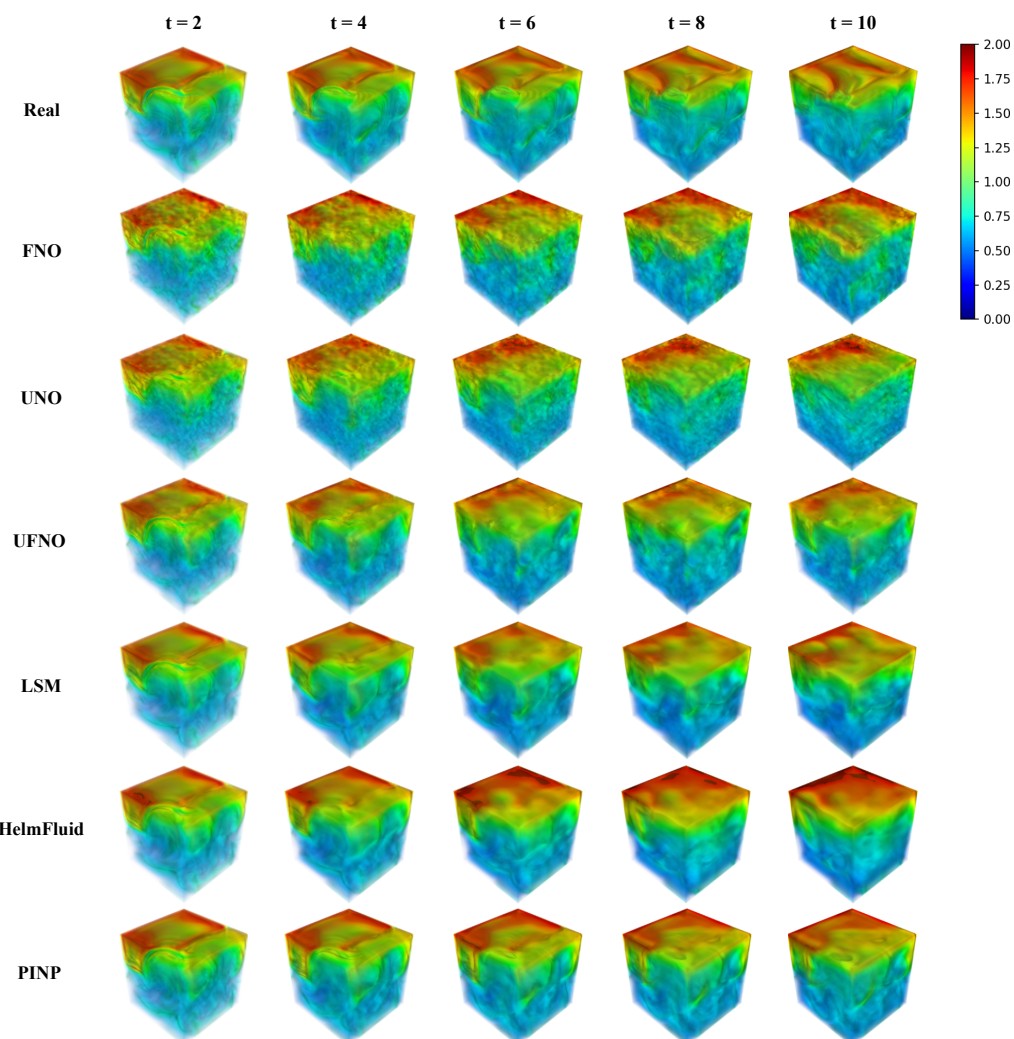

Figure S16: An example comparing our model with the baseline on 3D smoke data.

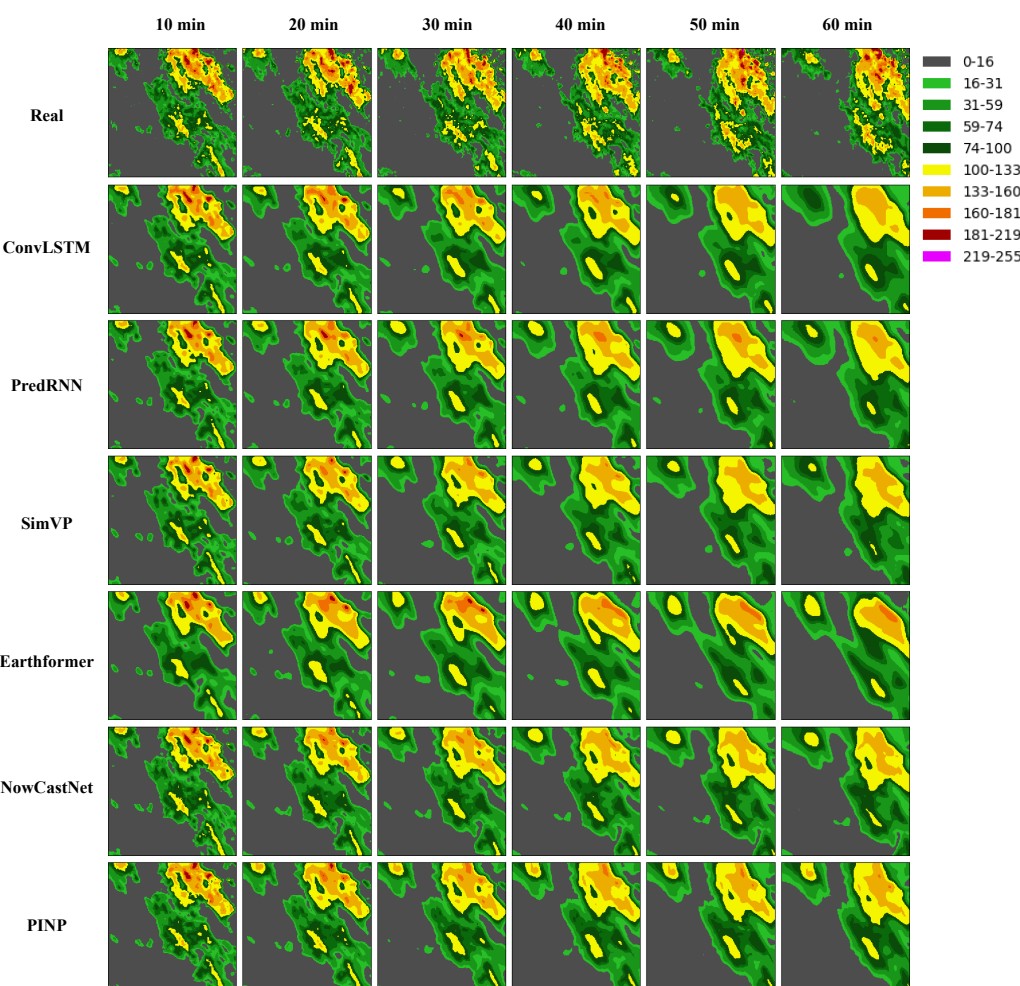

Figure S17: An example comparing our model with other baselines in nowcasting.

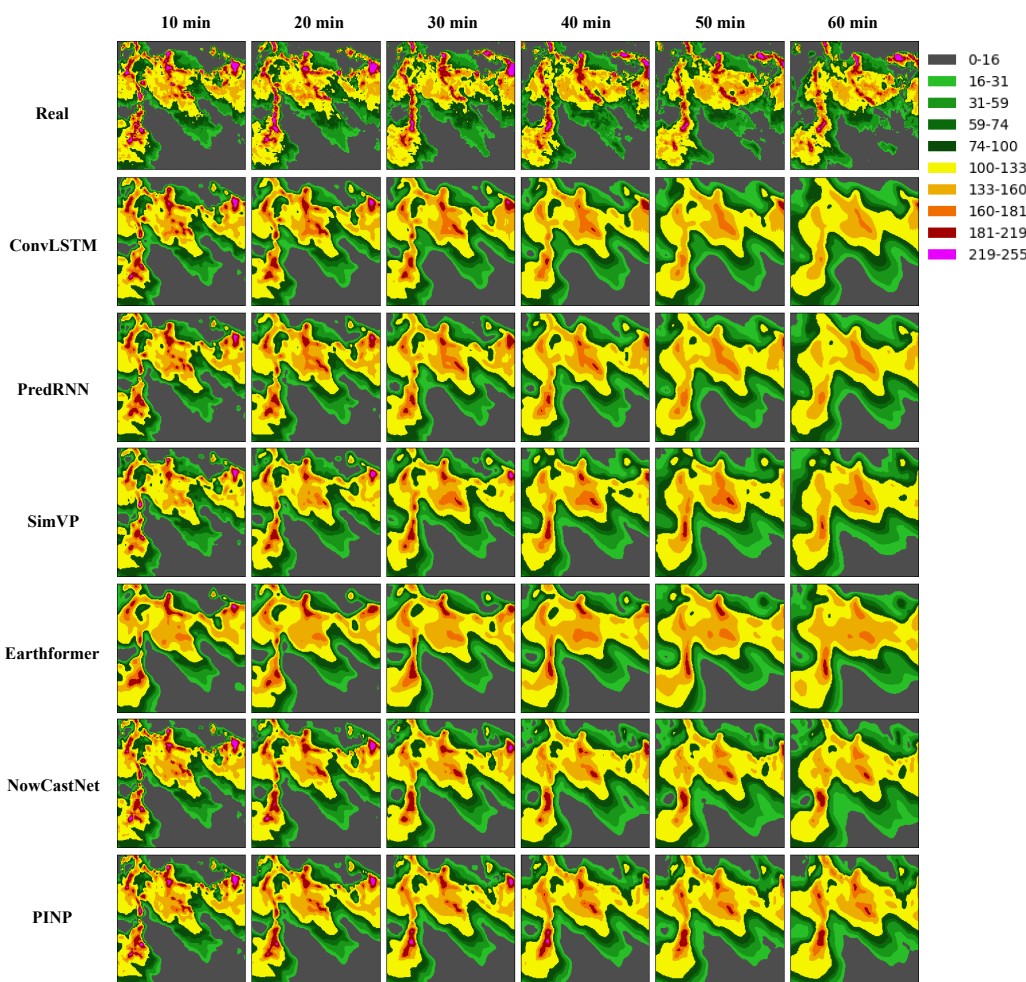

Figure S18: An example comparing our model with other baselines in nowcasting.

