# OpenReview forum: "PINP: Physics-Informed Neural Predictor with latent estimation of fluid flows"
_ICLR.cc/2025/Conference — ICLR 2025 Poster_

### Official Review · Reviewer_3QEU · 2024-10-27

**Soundness:** 2
**Presentation:** 3
**Contribution:** 2
**Rating:** 6
**Confidence:** 4

**Summary:**

This paper proposes a new physics-informed fluid predictor, named PINP. PINP firstly estimates the underlying pressure and velocity filed from observed fluid, which is constrained by a discretized physics loss. Then it employs an interpolation formalization of integral for future prediction, where an additional correction network is presented to reduce the error of discretized PDE predictor. Experimentally, PINP performs well in 2D and 3D flows and weather prediction tasks.

**Strengths:**

This paper is overall well-written.

The idea of incorporating physics loss into fluid prediction is reasonable.

The authors have provided comprehensive experiments to verify the effectiveness of the proposed method.

**Weaknesses:**

1.	The title is kind of overclaimed.


Since this paper is tailored to fluid prediction, I think “physics-informed fluid predictor” is more suitable. Otherwise, it is a little bit overclaimed, where there are extensive prediction tasks that PINP cannot solve, such as rigid body movement (controlled by classical physics) or magnetic field (governed by electromagnetism).

2.	A series of technical designs are underexplored or not well supported.

(1)	PINP adopts the discretized PDE loss for physics constraint, which may bring serious approximation error. The current design is based on the assumption that the differential operator can be approximated by spatial or temporal difference, which cannot be satisfied, especially in low-resolution data. Note that I am not saying that being physics-informed is a bad idea. The canonical physics-informed neural works employ the auto differential in neural works for approximation, which is much more precise than the discretization in PINP.

(2)	I cannot figure out that why additionally predicting the pressure field can boost the performance. As shown in Figure 2 (b), the predicted pressure field is only used in physical constraint loss, which cannot affect the future prediction process. This means that predicting the pressure field is just to fit the physical loss, which brings a new meaningless task. According to my experience, I think this design can only bring extra load to the model instead of benefiting the prediction. Besides, as shown in Figure 9(a), removing physical constraints will not bring a serious decrease. Further, How about keeping the second equation in Eq.(12) but removing the pressure-related one? I believe that the benefit of physics loss is mainly brought by the incompressible term loss.

(3)	The design of the correction network is also weird. As formalized in Eq.(10), the inputs and outputs of the correction network are both expected to be close to the ground truth. Under this constraint, why correction network is necessary? (Minor: Eq.(10) may have a typo, where the comma should be “-”).

(4)	About the temporal loss. I am curious about how likely is this loss function to work. Some statistical results on how many times this loss is non-zero are expected.

Going further from (2), I doubt that the prediction of pressure field is useless in the current design, which is listed as one of the main contributions w.r.t. other papers. I think compared with Helmfluid, the advantage of PNIP lies in the physical loss, which can provide a more direct and explicit constraint to the velocity field.

In summary, I think there are many unsupported designs in the proposed method, which may affect the claim of the main contribution of this paper.

3.	About the efficiency.

I am curious about the training overload. Since the calculation of loss in Eq.(16) may also cause extra computation costs than other baselines.

**Questions:**

1.	About implementation of baselines.

In NowCastNet, ensuring eidetic prediction results is one significant contribution of this paper. However, as shown in Figure 8, its prediction is quite blurry. I am wondering how the authors experimented with this baseline.

Besides, in the supplementary materials, the prediction results of LSM and FNO appear strange periodic shakes. Actually, I think a well-trained deep model will not make such weird predictions. Did the authors carefully tune these two baselines?

2.	About spatial generalization.

Why PINP can achieve spatial generalization? Can the authors provide some intuitive explanations?

---

> ### Author Response · Authors · 2024-11-21
> **Reply to Reviewer 3QEU (Part 1)**
>
> Thank you for your insightful comments and suggestions, which are very helpful in improving the clarity of our paper!
>
> > **Q1: The prediction blur issue in NowcastNet.**
>
> **Reply**: Excellent remark! NowcastNet can be divided into two components: the deterministic prediction model (Evolution Network) and the probabilistic generative model (GANs). The clarity of the predictions stems from the testing phase, where the "nearest" mode wrapping and the probabilistic generative model are applied. The experimental results for the Evolution Network alone and the complete NowcastNet are summarized below in **Table A**.
>
> **Table A: Comparison of NowCastNet results under different settings with Our Model**
> | Experiment                           | MSE(1e-3) | CSI-M  | CSI-219 | CSI-181 | CSI-160 | CSI-133 | CSI-74  | CSI-16  |
> |--------------------------------------|-----------|--------|---------|---------|---------|---------|---------|---------|
> | Evolution Network (wrap mode='bilinear') | **3.8883**  | 0.3743 | 0.0803  | 0.1811  | 0.2351 | 0.3515 | **0.6635** | **0.7339** |
> | Evolution Network (wrap mode='nearest')  | 6.1783    | 0.3529 | 0.0971 | 0.1858 | 0.2220  | 0.3110  | 0.5986  | 0.7027  |
> | NowcastNet                           | 5.7999    | 0.3549 | 0.0933  | 0.1844  | 0.2179  | 0.3104  | 0.6131  | 0.7106  |
> | PINP (our work)                      | 4.1684  | **0.3800** | **0.0989** | **0.1952** | **0.2448** | **0.3573** | 0.6556  | 0.7279  |
>
> It is crucial to note that there are two main approaches for nowcasting precipitation prediction:
>
> - Deterministic prediction models.
> - Enhancing deterministic models with probabilistic generative models.
>
> Since our model is deterministic (similar to the Evolution Network), we selected the Evolution Network's better results in our comparison. While incorporating a probabilistic model might improve visual results, it could negatively impact evaluation metrics. We believe presenting NowcastNet's results would not fairly represent the model's contributions.
>
> The above additional explanations have been included in the main text (Line 429-431, page 8, Section 4.2), and the NowcastNet results are provided in the Appendix A.5 (page 15).
>
> > **Q2: The issue of periodic oscillation in baseline results.**
>
> **Reply**: The periodic "shaking" phenomenon in some baseline model predictions is a form of periodic oscillation. Initially, this puzzled us as we did not find any bug in the implementation code. Upon further investigation, we discovered that the oscillation period (4 steps) matches the training rollout step size. Specifically, our training uses 4 frames to predict one frame unrolling over four steps, while the testing (inference) extends the rollout process to 40 steps. This periodic oscillation may arise because some models (except ours) struggle to learn motion dynamics within short sequence training.
>
> To further investigate this phenomenon, we conducted three validation experiments as follows:
>
> - Case 1: Training with four frames predicting one frame for four steps.
> - Case 2: The same as Case 1 but with an adjusted batch size.
> - Case 3: Training with four frames predicting one frame for six steps.
>
> Our findings include:
>
> - Experiments on Case 1 and Case 2 exhibited oscillations with a period of 4 throughout training (even at epoch 200).
> - Experiment on Case 3 initially had oscillations with a period of 6, but these gradually diminished as training progressed.
>
> This suggests that some baseline models struggle to capture long-term motion features when trained on short-term sequences. Under the configuration of Case 3, the oscillations in LSM were significantly reduced, yet our model still outperformed it, as shown in **Table B**.
>
> **Table B: Comparison of Results Between Our Method and LSM Under Experimental Setting Case 3.**
> | Method      | MAE   | MSE    |
> |-------------|-------|--------|
> | LSM         | 0.0099| 0.0002 |
> | PINP (ours) | 0.0062| 0.0001 |
>
> The details of the above analysis and results are included in our paper (page 7, Section 4.1 and page 30, Appendix F).
>
> > **Q3: Spatial generalization.**
>
> **Reply**: In our tests, the spatial generalization was demonstrated by adding/removing obstacles or altering their shapes—variations not present in the training data. The model's ability to generalize spatially arises from:
>
> - The Signed Distance Field (SDF), which effectively captures spatial information.
> - The explicit embedding of Navier-Stokes equations into the model and loss function, which enhances the learning of physical dynamics.
>
> For instance, the model learns that higher pressure in front of obstacles constrains velocity, resulting in spatial generalization in fluid flows.

---

> ### Author Response · Authors · 2024-11-21
> **Reply to Reviewer 3QEU (Part 2)**
>
> > **Weakness 1: Overclaimed title.**
>
> **Reply**:  Excellent Remark! We have taken your suggestion into account. Finally, We have revised the title to better reflect the paper's contributions:
> **"PINP: Physics-Informed Neural Predictor with Latent Field Estimator of Fluid Flows"**.
>
> > **Weakness 2a: Accuracy at low resolution.**
>
> **Reply**:  In our experiments, we used 3D scenes with a resolution of 64×64×64, which is not high. Under the same discretization method, our approach consistently outperformed all baselines.
>
> We acknowledge that discretization introduces errors, which is why we designed a correction network to mitigate them. Physics-informed neural networks (PINNs) using automatic differentiation can achieve higher precision but are more suited to MLP-based designs, which can be less effective for comprehensive predictive tasks, particularly for complex or long-term predictions due to limited representation ability.
>
> > **Weakness 2b: Necessity of predicting pressure fields and its ablation study.**
>
> **Reply**: The predicted pressure field serves not only as an auxiliary feature for predictions but also provides interpretability. Its similarity to actual distributions aids in understanding the dynamics, especially around obstacles or boundaries. In our paper, this constraint is written as:
>
> $$\boldsymbol{e_1}(\boldsymbol{x},t_k)=\Delta \boldsymbol{u}(\boldsymbol{x},t_k) + \left(-\boldsymbol{u}(\boldsymbol{x},t_k) \cdot \nabla \boldsymbol{u}(\boldsymbol{x},t_k) - \nabla p(\boldsymbol{x},t_k) + \text{Re}^{-1}\nabla^2 \boldsymbol{u}(\boldsymbol{x},t_k)\right) \Delta t.$$
>
> The constraint we introduce can actually be viewed as a temporal constraint on the velocity field ($\Delta \boldsymbol{u}( \boldsymbol{x},t_k)$ denotes the difference in velocity between consecutive frames). The inclusion of this temporal constraint helps stabilize the velocity field over time, which is crucial for long-term predictions.
>
> Following your suggestion, we designed an additional ablation study by removing $\boldsymbol{e_1}$, referred to as **no Velocity-Pressure Constraint**. The results are summarized below in **Table C**. We have also revised the paper accordingly (Table 4, page 9, Section 5 and Appendix D.1, page 19).
>
> **Table C. Comparison of Evaluation Metrics Between Different Constraint Configurations and the Original Method.**
> | Setting                     | MAE   | MSE    |
> |-----------------------------|-------|--------|
> | no Physical Constraint      | 0.0272| 0.0089 |
> | no Temporal Constraint      | 0.0293| 0.0108 |
> | no Velocity-Pressure Constraint | 0.0262| 0.0087 |
> | Normal                      | 0.0209| 0.0057 |
>
> In addtion, the result presented in Figure 9(a) reflects an average outcome across all 2D fluid test cases. The 40-step extrapolation limit made it challenging to illustrate the long-term impact of different components. To better illustrate the effectiveness of the Physical Constraint, we visualized the results on the Smoke2D dataset with 100-step extrapolation, which better highlights such an effectiveness. We have updated the paper accordingly (Figure 9(a), page 10, Section 5).
>
> > **Weakness 2c: Necessity of the correction network.**
>
> **Reply**: Our predictions rely on the following equation:
>
> $$\hat{c}'(\boldsymbol{x},t_{k+1}) = c(\boldsymbol{x},t_k) + \left(-\boldsymbol{u}(\boldsymbol{x},t_k) \cdot \nabla c(\boldsymbol{x},t_k) + \text{Pe}^{-1} \nabla^2 c(\boldsymbol{x},t_k)\right) \Delta t.$$
>
> The use of second-order differences to approximate $\nabla$ introduces unavoidable errors. The correction network overcomes these limitations by minimizing these errors. Without it, the model struggles with long-term stability. The quantitative comparisons highlight the correction network's significance (Table 4, page 9, Section 5), with the results also shown in **Table D**.
>
> **Table D: Comparison of our model’s results with those obtained after removing the Correction Network on 2D smoke data.**
> | Setting             | MAE   | MSE    |
> |---------------------|-------|--------|
> | no Correction Network | 0.0316| 0.0098 |
> | Normal             | 0.0209| 0.0057 |
>
> > **Weakness 2d: How likely is the Temporal Constraint function to work?**
>
> **Reply**: Great question! In Appendix C.2, we have provided an analysis of the temporal loss variation during training. To further clarify the role of the Temporal Constraint, we have updated the presentation in Figure S5 (right) (page 18, Appendix C.2) of the revised paper to improve intuitiveness.
>
> Here is a summary of the results:
>
> - At the beginning of training, the temporal loss is non-zero.
> - As training progresses, the probability of the temporal loss approaching zero increases. However, the temporal loss does not become consistently zero.
>
> Hence, we can conclude that the Temporal Constraint works  and plays an important role in improving the model's prediction accuracy.

---

> ### Author Response · Authors · 2024-11-21
> **Reply to Reviewer 3QEU (Part 3)**
>
> > **Weakness 3: About the efficiency.**
>
> **Reply**: We agree that the inclusion of the temporal constraint and additional loss functions requires higher computational resources, particularly GPU memory. However, the impact is manageable and comparable to other advanced baselines. For example, in the 3D scenario, the GPU memory consumption of PINP is on par with Helmfluid as shown in **Table E**.
>
> **Table E: Comparison of GPU Memory Consumption Between Our Model and HelmFluid on 3D Smoke Data.**
> | Method        | GPU Memory Consumption (Mb) |
> |---------------|------------------------------|
> | HelmFluid     | 22,892                       |
> | PINP (this work) | 21,744                       |
>
> In summary, we appreciate your constructive comments and suggestions. Please let us know if you have any other questions. We look forward to your feedback!

---

> ### Author Response · Authors · 2024-11-23
> **Looking forward to your feedback**
>
> Dear Reviewer 3QEU,
>
> Again, thanks for your constructive comments. We would like to follow up on our rebuttal to ensure that all concerns have been adequately addressed. If there are any further questions or points that need discussion, we will be happy to address them. Your feedback is invaluable in helping us improve our work, and we eagerly await your response.
>
> Thank you very much for your time and consideration.
>
> Best regards,
>
> The Authors

---

> > ### Comment · Reviewer_3QEU · 2024-11-25
> >
> > I want to thank the authors's great effort in clarifying experiment settings and adding ablations.
> >
> > (1) My concerns about baselines are all resolved. One more suggestion about the current experiments: I think training models with "four frames predicting one frame for six steps" is a more reasonable way to examine the model performance, which can ensure the model is well adapted to the long-term evaluation.
> >
> > (2) My questions about spatial generalization are clear. However, I still cannot understand why predicting pressure can benefit the final performance. Note that I acknowledge the advantage of PINP in interpretability. If you do not apply additional supervision on pressure and the predicted pressure is not used for generating the final prediction, how does this additional output help final performance? In my opinion, this branch is more like an auxiliary task that constrains the output pressure to satisfy physical constraints for "interpretability".
> >
> > Overall, I appreciate this paper's performance and sufficient experiments, while some designs are over-complicated (such as temporal consistent loss and correction networks). Thus, I will raise my score to 5 but cannot give a position rating for the current design.

---

> ### Author Response · Authors · 2024-11-25
> **Clarification on Reviewer 3QEU's Question (Part1)**
>
> >**Q1: The Role of Pressure Fields in Assisting Concentration Prediction.**
>
> **Reply**: Thank you for this insightful comment. We believe that the pressure field $( p )$ aids concentration prediction by effectively constraining the velocity field $\boldsymbol{u}$, which, in turn, improves the prediction of concentration. We formalize this constraint as the **Velocity-Pressure Constraint**, defined by the following equation:
>
> $$\boldsymbol{e_1}(\boldsymbol{x},t_k)=\Delta \boldsymbol{u}(\boldsymbol{x},t_k) + \left(-\boldsymbol{u}(\boldsymbol{x},t_k) \cdot \nabla \boldsymbol{u}(\boldsymbol{x},t_k) - \nabla p(\boldsymbol{x},t_k) + \text{Re}^{-1}\nabla^2 \boldsymbol{u}(\boldsymbol{x},t_k)\right) \Delta t.$$
>
> In this equation, by introducing $H \times W + 1$ additional unknowns ( $p(\boldsymbol{x},t_k) \in \mathbb{R}^{H \times W}$ and $\text{Re} \in \mathbb{R}$ ), we impose $2 \times H \times W$ constraints ( $\boldsymbol{e_1} \in \mathbb{R}^{2 \times H \times W}$ ). This improves the velocity field's accuracy, which positively impacts concentration predictions.
>
> Referring to [1], we evaluated the correlation between predicted and true latent physical quantities, with and without the Velocity-Pressure Constraint. The results, summarized in **Table A**, show a clear improvement in the correlation of velocity components when the constraint is applied:
>
> **Table A: Correlation Between Predicted and True  Latent Physical Quantities**
> | Latent Physical Quantities | Correlation (Normal) | Correlation (No Velocity-Pressure Constraint) |
> |----------------------------|----------------------|-----------------------------------------------|
> | $u_x$                 | 0.9008              | 0.8611                                        |
> | $u_y$                 | 0.9118              | 0.8647                                        |
> | $p$                   | 0.8965              | -                                             |
>
> Figure S6 (page 20, Section D.1) provides a visual comparison of velocity fields with and without the constraint.
>
> For concentration prediction, **Table B** compares evaluation metrics across different settings, including configurations without the Velocity-Pressure Constraint, Temporal Constraint, or Correction Network.
>
> **Table B: Comparison of Evaluation Metrics Across Different Settings**
> | Setting                    | MAE    | MSE    |
> |----------------------------|--------|--------|
> | No Correction Network      | 0.0316 | 0.0098 |
> | No Temporal Constraint     | 0.0293 | 0.0108 |
> | No Velocity-Pressure Constraint | 0.0262 | 0.0087 |
> | Normal                     | 0.0209 | 0.0057 |
>
> The results demonstrate that introducing the Velocity-Pressure Constraint significantly improves concentration prediction accuracy.
>
> In summary, we believe that the pressure field $p$ not only enhances interpretability but also contributes to improving concentration prediction.
>
> ***Reference:***
>
> [1] Kochkov, et al. Machine learning–accelerated computational fluid dynamics. PNAS, 2021, 118.21: e2101784118.

---

> ### Author Response · Authors · 2024-11-25
> **Clarification on Reviewer 3QEU's Question (Part2)**
>
> >**Q2: Concerns About Over-Complicated Designs (Temporal Consistent Loss and Correction Networks).**
>
> **Reply**: Regarding the impact of the Correction Network and Temporal Constraint, we have clarified their contributions in **Table B**, where both modules show clear benefits for prediction accuracy. Below, we explain the motivations for these components in detail.
>
> **Motivation for the Temporal Constraint**
>
> The prediction task can be expressed as:
> \begin{equation}
>   c(\boldsymbol{x},t_{k+1})=\int_{t_k}^{t_{k+1}} \dot{c}(\boldsymbol{x},t) dt + c(\boldsymbol{x},t_k) \tag{a}
> \end{equation}
>
> Directly solving this integral is challenging. By leveraging the Lagrange Mean Value Theorem and assuming material concentration $c(\boldsymbol{x}, t)$ is continuous, there exists a moment $t' \in [t_k, t_{k+1}]$ such that:
>
> \begin{equation}
>    {\int_{t_k}^{t_{k+1}} \dot{c}(\boldsymbol{x},t) dt = c(\boldsymbol{x},t') \Delta t, \ t' \in[t_k, t_{k+1}].\tag{b}}
>    \end{equation}
>
> Thus, the prediction can be reformulated as:
>
> \begin{equation}
>    \begin{split}
>    c(\boldsymbol{x},t_{k+1}) &= c(\boldsymbol{x},t_k) + c(\boldsymbol{x},t') \Delta t\\
>    &= c(\boldsymbol{x},t_k)+\left(-\boldsymbol{u}(\boldsymbol{x},t')\cdot \nabla c(\boldsymbol{x},t') +\text{Pe}^{-1} \nabla^2 c(\boldsymbol{x},t')\right) \Delta t .
>    \end{split} \tag{c}
>    \end{equation}
>
> While $\boldsymbol{u}(\boldsymbol{x}, t')$ can be modeled using a neural network, handling $c(\boldsymbol{x}, t')$ requires further consideration. We explored two approaches:
>
> - Approximate $c(\boldsymbol{x}, t')$ with $c(\boldsymbol{x}, t_k)$.
> - Use a neural network to map past concentrations to $c(\boldsymbol{x}, t')$.
>
> The first approach performed well for short-term predictions but deteriorated rapidly in long-term extrapolation (Figure 9\(c\), page 10, Section 5) due to accumulated errors in velocity fields (Figure S8, page21, Section D.3).
>
> Therefore, we adopted the second approach, introducing **Temporal Constraints** to ensure physical meaning of $c(\boldsymbol{x}, t')$ and $\boldsymbol{u}(\boldsymbol{x}, t')$. Because in Equation \(c), $c(\boldsymbol{x}, t')$ is coupled with $\boldsymbol{u}(\boldsymbol{x}, t')$. Without these constraints, $c(\boldsymbol{x}, t')$ would lose its physical meaning, which would in turn cause $\boldsymbol{u}(\boldsymbol{x}, t')$ to lose its physical validity (Figure S6, page 20, Section D.1), ultimately negatively impacting long-term predictions (Figure 9\(a\), page 10, Section 5).
>
>
> **Motivation for the Correction Network**
>
> Equation \(c\) approximates gradient operators with second-order differences, which introduces inherent errors. Without a Correction Network, these errors persist in the final prediction step, limiting the model's performance, particularly in long-term forecasting.
>
> For instance, predictions results in manageable errors during short-term testing. However, when tested on long-term, error accumulation becomes severe. As illustrated in Figure 9(b) (page 10, Section 5), while both models perform similarly within 40 frames, the model with the Correction Network significantly outperforms in longer sequences. Figure S7 (page 21, Section D.2) shows visual comparisons, highlighting the Correction Network's importance.
>
> Given the challenges of long-term sequence prediction, we conclude that the Correction Network is essential for mitigating accumulated errors.
>
> In summary, we believe that the introduction of the Correction Network and Temporal Constraint is essential. The Temporal Constraint is a natural requirement of the method we propose, while the Correction Network is designed to overcome the model bottleneck caused by errors introduced by central difference methods in the experiments. The Temporal Constraint helps preserve physical validity, and together with the Correction Network, it improves the performance of long-term predictions.
>
> Thank you very much for your great questions! Please let us know if you have any other questions. Looking forward to your response!

---

> > ### Comment · Reviewer_3QEU · 2024-11-26
> >
> > Thanks for your further clarification. The results of Pressure loss are intuitive. I decided to raise my score to 6.
> >
> > Again, I appreciate the strong experiments in this paper. The overall design is reasonable. However, I am not claiming that the correction networks or temporal consistent loss are useless, what I want to express is that these plug-in modules just made the overall design more complicated. The design is just reasonable but not neat.

---

> > > ### Author Response · Authors · 2024-11-26
> > > **Thanks for your positive feedback**
> > >
> > > Dear Reviewer 3QEU,
> > >
> > > Thank you very much for your positive and encouraging feedback. Your constructive comments, as well as time and effort placed on reviewing our paper, are highly appreciated!
> > >
> > > Best regards,
> > >
> > > The Authors

---

### Official Review · Reviewer_tKs8 · 2024-10-29

**Soundness:** 4
**Presentation:** 3
**Contribution:** 3
**Rating:** 8
**Confidence:** 3

**Summary:**

This paper's core idea is to combine a data-centric deep learning approach with physics by incorporating the discretised Navier-Stokes equations into the neural network architecture and constraining the loss function. By explicitly incorporating the governing equations and the associated physical quantities, the authors try to model the system and help with consistency, interpretability, and extrapolation capabilities. The extensive proposed experiments show good performances and generalisation to unseen domains.

**Strengths:**

- The paper is very well written and explains complex problems clearly.
- While the idea of incorporating PDE-based constraints in the network architecture and loss function is not new, the author presents a set of methods, tricks, and ideas that make the technique work better than previous literature.

**Weaknesses:**

- Interpretability depends on the other physical quantities' models. While there are theoretical reasons to believe the quantities are interpretable, there is little experimental evidence.
- The pertinence of benchmarking nowcasting and the advantage of this method over other neural operator-based methods for this task is unclear.

**Questions:**

- The gradient discretisation used here is a second-order central difference approximation. Could you elaborate on whether there were specific architectural reasons for choosing this method over other discretisation schemes? Additionally, it would be helpful to understand if you considered alternative discretisation approaches
You compare two different sets of baseline: one for now-casting and one for Navier-Stoke simulation. Why is your model capable of doing both, as other neural operator-based models are not? That seems like a significant advantage that has yet to be developed.

---

> ### Author Response · Authors · 2024-11-21
> **Reply to Reviewer tKs8**
>
> Thank you for your constructive comments and suggestions!
>
> > **Q1. The gradient discretization used here is a second-order central difference approximation.**
>
> **Reply**: Thank you for your insightful comment! Your feedback helps us clarify the approach used in our method.
>
> - **Automatic Differentiation and Central Difference.** Physics-Informed Neural Networks (PINNs) typically use automatic differentiation within neural networks to achieve high accuracy in approximations. However, this differentiation method relies on the MLP (multi-layer perceptron) network structure, which is typically less effective for comprehensive predictive tasks, particularly for complex or long-term predictions, due to its limited representation ability.
>
> - **Second-order Central Difference and Higher-order Central Difference.** In our experiments, we primarily use the second-order central difference approximation. In non-boundary regions, higher-order central differences did not show a significant advantage in our model's prediction accuracy. However, in boundary regions, using more points in higher-order differences often led to a larger influence from zero-value boundaries, which adversely affected the prediction accuracy. This impact was especially notable in long-term predictions, where error accumulation became more significant. Hence, we employ the second-order central difference considering the above tradeoff.
>
> > **Q2. Two Different Sets of Baselines.**
>
> **Reply**: Thank you for this comment. In nowcasting, we treat it as the transport of rain clouds by atmospheric motion, but compared to simulated data, it presents unique challenges. These challenges include the lack of strict adherence to the Navier-Stokes (NS) equations and the presence of significant noise. Most current neural operator methods are tested with noise-free or low-noise simulated data.
>
> In our method, we discretize the Partial Differential Equations (PDEs) and embed them into our model and loss function. Although the precipitation process does not strictly follow the NS equations, we believe the dynamics described by the NS equations still captures some essential motion features that benefit the prediction. By using nowcasting precipitation, we test our model’s ability to handle scenarios with noisy data and partial adherence to physical equations.
>
> > **Weakness 1. About Interpretability.**
>
> **Reply**: Thanks for your concern regarding interpretability. However, we believe that providing unsupervised predictions of the velocity and pressure fields based solely on observed concentration data is physically meaningful. This aids in understanding the future dynamics.
>
> > **Weakness 2. The pertinence of benchmarking nowcasting.**
>
> **Reply**: Excellent Remark! As previously mentioned, the SEVIR test is designed to assess the predictive capability of our model under conditions of noisy data and in scenarios where the governing equations do not strictly satisfied. In addition, we tested the most representative neural operator, namely, FNO, for nowcasting. Unfortunately, its performance is unsatisfactory given the SEVIR dataset. Hence, we decide not to include such results in our paper.

---

### Official Review · Reviewer_m8WU · 2024-10-30

**Soundness:** 3
**Presentation:** 2
**Contribution:** 3
**Rating:** 6
**Confidence:** 5

**Summary:**

This work presents a new physics-informed learning approach that enables the prediction of coupled physical states, under a partially observed data environment. It applies the discretization of physical equations, integrating into the model architecture and loss function. The superior performance is shown in four benchmarks including a real-world data.

**Strengths:**

1. The proposed method enhances the simulation capacity of PDEs, especially on the long-term prediction.
2. The proposed method is tested on multiple benchmarks across different scenarios, especially on the real-world measured dataset.
3. The authors vividly demonstrated the simulation process through videos.

**Weaknesses:**

1. The reviewer believes that there are significant issues with the introduction of the method. The method is complex and lacks an overview of the proposed method. The reviewer understands that the proposed method first outputs $p(t')$, $u(t')$, and $c(t')$ through a physical inference network, where these three outputs are constrained by physical and temporal conditions. Then $\hat{c}'(t+1)$ is computed through discretized PDEs, after which $\hat{c}'(t+1)$ and $c(t)$ are fed into another network for prediction, while simultaneously incorporating a data loss with the label. If this understanding is correct, the reviewer questions the novelty of this work, as it merely sandwiches numerical FDM calculations between neural networks. Moreover, the motivation for this approach remains unclear.
2. In line 039, the statement is inaccurate and needs to be referenced to the literature. The reviewer points out that velocity fields can be observed through techniques such as PIV and PTV.
3. In line 069, what does the past observable data mean. Authors should introduce more about the settings.
4. In line 102, "often difficult to obtain in practical applications". The reviewer considers the statement is inappropriate, as initial conditions are typically obtainable when solving PDEs.
5. In Table 1, the reviewer appears to have misinterpreted the meaning of the three categories in this table. If 'velocity' refers to velocity fields, then this table is not appropriate, as FNO (Fourier Neural Operator) is equally capable of predicting both velocity and pressure fields.
6. In Eqn. 3, why does this equation still integrate from t to t+1? A more detailed derivation process is needed to help readers understand. This is crucial for comprehending the motivation behind the problem. What is the meaning of $\Delta t$?
7. In Sec. 3.4, the introduction is oversimplified, merely stating which networks are used. This raises two concerns for the reviewer: first, why was U-Net chosen over more advanced transformer architectures, and second, too many network structural details are omitted, forcing readers to consult the appendix for understanding.
8. In Sec. 3.5, author should carfully introduce the training process as there are many networks and parameters. Are they trained in an end-to-end manner? This raises a question about how the physials inference network can simultaneously learn Pe and output flow fields. These two components might interfere with each other, potentially making the network untrainable. Has the use of stop-gradient operations like VQVAE been considered?
9. What is the PDE for the real-world data? Is it explicitly known? Real data often comes with noise - has this method considered noise effects, or are there any approaches proposed to address the influence of noise?
10.  In Sec. 5, especially in Fig. 9(a), the authors need to specify the number of experimental trials conducted and report the confidence levels, as it appears that the two constraints overlap for an extended period of time.
11. What is the detail setting of the fluid 2D data, including $\nu$, $dx$, $dt$, and boundary conditions?
12. The reviewer does not find the link of code and dataset from the paper. Code and data are important criteria for verifying the rationality of results. Will the author make them opensource？

**Questions:**

Please check the weaknesses.

---

> ### Author Response · Authors · 2024-11-21
> **Reply to Reviewer m8WU (Part 1)**
>
> Thank you for your constructive comments and suggestions!
>
> Before addressing the specific questions, we would like to clarify the primary problem we are motivated to solve as follows:
>
> *We consider partially observed data environments, e.g., the rising and diffusion of smoke in the air. In such scenarios, while smoke concentration can be easily observed (by video cameras), the velocity and pressure fields are not directly observable. Our goal is to predict future smoke concentration while simultaneously estimating velocity and pressure fields as interpretable evidence in an unsupervised manner. To align with potential real-world applications, our 2D simulated data consists of grayscale image sequences where even the spatial scale remains unknown.*
>
> > **Weakness 1. Lack of a clear overview of the method; the approach is an extension of FDM which lacks novelty.**
>
> **Reply**: Thanks for your comments! We acknowledge that our original overview of the proposed method was overly brief. This section has been thoroughly revised, and the updated version is now presented in lines 153–188 (pages 3–4, Section 3) in the revised paper.
>
> Unlike numerical methods such as the Finite Difference Method (FDM), our approach leverages the temporal continuity of concentration and applies the Lagrange Mean Value Theorem for temporal discretization. This embedding strategy, coupled with our loss function design, enables the model to provide interpretable estimation of velocity and pressure fields, without the need of training data, while maintaining stability in long-term predictions.
>
> Moreover, we emphasize that our contribution extends beyond achieving high prediction accuracy over longer time horizons. A key novelty lies in our ability to infer hidden physical quantities (e.g., velocity and pressure) in an unsupervised manner. These quantities closely resemble actual physical fields and offer interpretability, helping to enhance the understanding of future motion processes. This forms fundamental differences compared with FDM.
>
> > **Weakness 2. Inaccuracy in claims about velocity field observability.**
>
> **Reply**: Great remark! Our initial phrasing was indeed overly absolute, and we have revised it in lines 40–41 (page 1, Section 1) of the paper. Please also see below.
>
> *"Successful approaches usually rely on supervised learning for these quantities, but such data is not easy to obtain in practical scenarios.""*
>
> Nevertheless, while velocity fields can be observed using methods like Particle Image Velocimetry (PIV) and Particle Tracking Velocimetry (PTV), these methods face notable challenges, such as particle occlusion, high experimental costs, limited spatial resolution, etc.
>
> > **Weakness 3. Ambiguity in terms like "past observable data" .**
>
> **Reply**: Excellent comment! Our task indeed involves partially observed data environments, where only the concentration field is directly observable. We infer the underlying mapping based on the past observations of concentration data. To better reflect this, we have revised the phrasing in line 69 (page 2, Section 1) of the paper. Please also see below.
>
> *"Neural networks are employed to establish a mapping function from past measurement (observed data) to both intermediate observed quantities and latent physical variables, enabling predictions through the discretized NS equations."*
>
> > **Weakness 4. Inappropriate statement about the difficulty of obtaining initial conditions.**
>
> **Reply**: Obtaining explicit initial conditions is often impractical in real-world scenarios. For instance, when observing the rising of smoke, it is far more common to have access to a video rather than explicit initial conditions. This is why our 2D simulation data is structured as grayscale image sequences, with the actual spatial size remaining unknown.
>
> > **Weakness 5. Misinterpretation of Table 1 categories and the capabilities of FNO.**
>
> **Reply**: Thanks for your careful observation! While methods like FNO can map concentration to velocity fields, this mapping is supervised and requires velocity field data during training. ***Our approach, in contrast, predicts velocity fields using only concentration data, without requiring any ground-truth velocity fields for supervision.***
>
> Regarding Table 1, we have made modifications and added descriptive explanations (lines 147–149, page 3, Section 2).

---

> ### Author Response · Authors · 2024-11-21
> **Reply to Reviewer m8WU (Part 2)**
>
> > **Weakness 6. Need for detailed derivation of Equation 3.**
>
> **Reply**: Excellent remark! The proof relies on the **Lagrange Mean Value Theorem**, and we appreciate the suggestion to clarify this further. We have included additional explanations in the paper (lines 208-211, page 4, Section 3.1). Please see below.
>
> *"However, considering the continuity of material concentration $c(\boldsymbol{x},t)$ over time, based on the Lagrange Mean Value Theorem, there must exist a moment $t'$ between time $t_k$ and $t_{k+1}$ such that:"*
> \begin{equation}
>    \int_{t_k}^{t_{k+1}} \dot{c}(\boldsymbol{x},t) dt = c(\boldsymbol{x},t') \Delta t, \ t' \in[t_k, t_{k+1}].
>    \end{equation}
>
> Regarding the time notation $\Delta t$, we agree that our original description was imprecise. We have revised the formulations in the entire paper thoroughly. Specifically:
> - $k \in \{1, 2, \dots, N\}$ denotes the $k$-th frame.
> - $t_k$ represents the time of the $k$-th frame.
> - $\Delta t$ is the time interval between $t_k$ and $t_{k+1}$.
>
> > **Weakness 7. Justification for using U-Net over transformers.**
>
> **Reply**: The Transformer model typically consumes more GPU memory than the U-Net model. Due to page limitations, we could not provide detailed descriptions of the network architecture in the main text. Hence, we placed the detailed network architecture in Figure S3 (Appendix A.4, page 15). We hope for your understanding.
>
> > **Weakness 8. Lack of details on potential interference between components.**
>
> **Reply**: The model is trained in an end-to-end manner. We directly use the loss function we designed during training. Importantly, there is no interference between components in the actual training process. The process is straightforward and stable.
>
> > **Weakness 9. Treatment of noise in real-world data and explicitness of underlying PDEs.**
>
> **Reply**: Excellent comment! We would like to clarify that we herein use the Navier-Stokes (NS) equations as an approximation. While the atmosphere’s dynamics (e.g., raincloud transport) do not strictly follow NS equations, these equations effectively capture key motion patterns. This inductive bias is considered helpful to improve the model's robustness against noise.
>
> > **Weakness 10. Result in Figure 9(a) is unclear.**
>
> **Reply**: Thank you for pointing out the issue with Figure 9(a). This result reflects an average outcome across all 2D fluid test cases. The 40-step extrapolation limit made it challenging to illustrate the long-term impact of different components.
>
> To address this, we visualized the results on the Smoke2D dataset with 100-step extrapolation, which better highlights these effects. We have updated the results in the revised paper accordingly (Figure 9(a), page 10, Section 5). The results are also presented in **Table A** below.
>
> **Table A: Comparison of Evaluation Metrics Between Different Constraint Configurations and the Original Method.**
> | Ablation Setting            | MAE   | MSE   |
> |-----------------------------|-------|-------|
> | No Physical Constraint      | 0.0272| 0.0089|
> | No Temporal Constraint      | 0.0293| 0.0108|
> | No Velocity-Pressure Constraint | 0.0262| 0.0087|
> | Normal                      | 0.0209| 0.0057|
>
> > **Weakness 11. Insufficient details about the settings for 2D fluid data.**
>
> **Reply**: The symbols in the paper were deliberately chosen to avoid implying that these quantities are known. In our setup, the 2D data comprises grayscale image sequences (videos) of concentration, and the specific physical scales are unknown. We have added further clarifications and additional details in Figure S4 (page 16, Appendix B.1).
>
> > **Weakness 12. Absence of code and dataset links.**
>
> **Reply**: Open-sourcing the code and data is our priority. We will make both resources publicly available after the peer review stage.

---

> > ### Comment · Reviewer_m8WU · 2024-11-22
> > **Thank you for your response.**
> >
> > Thanks for your detailed responses. A part of my questions has been addressed by the responses. However, there is still an important issue that the author has not clarified, how are evaluation metrics calculated?
> >
> > This issue is important because if only the concentration is calculated without velocity and pressure. It is not considered a partial observation problem (partial observation refers to observation that is only partial but needs to be evaluated globally) for the quantitative contrast results in the table, and it will not be a limitation for methods such as FNO. However, if the calculation is based on all variables such as concentration, velocity, and pressure, the quantitative results in the table cannot be calculated for methods like FNO where velocity cannot be predicted (without labels). This seems to be a difficult problem to solve, and the author may need to redefine and present it more clearly. (In appendix B.2, authors only provided the formula without details.)
> >
> > Therefore, I will not change my score for now.

---

> > > ### Author Response · Authors · 2024-11-25
> > > **Request your feedback before the end of the discussion period**
> > >
> > > Dear Reviewer m8WU:
> > >
> > > As the author-reviewer discussion period will end soon, we would appreciate it if you could review our responses at your earliest convenience. If there are any further questions or comments, we will do our best to address them before the discussion period ends.
> > >
> > > Thank you very much for your time and efforts. Looking forward to your response!
> > >
> > > Sincerely,
> > >
> > > The Authors

---

> ### Author Response · Authors · 2024-11-22
> **Your concern on evaluation metrics has been clarified**
>
> >**Comment: Quantitative Comparison of Latent Physical Quantities.**
>
> **Reply**: Excellent comment! Referring to [1], we calculate the Correlation to quantitatively evaluate the accuracy of our predictions of the latent physical quantities compared to the true latent physical quantities. For example, we consider $x_i$ as the predicted values and $y_i$ as the true values, with $\overline{x}$ being the mean of all predicted values and $\overline{y}$ being the mean of all true values. The Correlation metric is defined as follows:
>
> \begin{equation}
> \mathrm{Correlation} = \frac{\sum_{i=1}^n\left(x_i-\bar{x}\right)\left(y_i-\bar{y}\right)}{\sqrt{\sum_{i=1}^n\left(x_i-\bar{x}\right) ^2} \sqrt{\sum_{i=1}^n\left(y_i-\bar{y}\right)^2}}
> \end{equation}
>
> The results of the Correlation metrics are shown in **Table A**.
>
> **Table A: The Correlation between the predicted latent physical quantities and the true values.**
> | Latent Physical Quantities           | Correlation  |
> |------------------|-------|
> | $u_x$ ($t$ = 50)      |0.9099 |
> | $u_y$ ($t$ = 50)  |0.9039|
> | $p$ ($t$ = 50) | 0.9251|
> | $u_x$ ($t$ = 100)       |0.9123 |
> | $u_y$ ($t$ = 100)      |0.9207|
> | $p$ ($t$ = 100) | 0.9061|
>
> We can see that high Correlation values (>0.9) in **Table A** further demonstrates the accuracy of the predicted hidden physical quantities. The above results have been updated in the latest version of our paper (Figure 6\(c\), page 8, Section 4.1 and Section B.2, page 17).
>
> Thank you very much for your great suggestion! Please let us know if you have any other questions.
>
> ***Reference:***
>
> [1] Kochkov, et al. Machine learning–accelerated computational fluid dynamics. PNAS, 2021, 118.21: e2101784118.

---

> > ### Comment · Reviewer_m8WU · 2024-11-26
> >
> > Thanks for your response.
> >
> > I have read the metrics and table in the manuscript, however the author may not understand my question. What I mean is that the FNO and other baselines cannot calculate the correlation metrics due to methods like FNO where velocity cannot be predicted (without labels as authors mentioned). It is hard draw the conclusion that the proposed method is superior only through the correlation metrics without comparing to baselines.
> >
> > Moreover, using the correlation following [1] in this work is not appropriate. [1] studied turbulence simulation, and due to the chaotic nature of turbulence, only correlation metrics can be considered. However, for the scenario in this manuscript, turbulence is not involved, so MSE should be able to better evaluate the prediction performance.

---

> > > ### Author Response · Authors · 2024-11-26
> > > **Reply to additional comment by Reviewer m8WU: clarification on the baseline comparison**
> > >
> > > Thank you for your valuable feedback. We would like to provide further clarifications regarding the concerns you raised.
> > >
> > > > **Comment 1: Clarification on the baseline comparison.**
> > >
> > > **Reply:** First, let us clarify the problem we are tackling: *Our goal is develop a generalizable model to predict concentration of a scalar field (e.g., smoke, dye, precipitation, etc.) while simultaneously estimating velocity and pressure fields (without any training labels) as interpretable evidence in an unsupervised manner.* For example, in real-world scenarios such as rising smoke, recorded videos of the smoke concentration are more readily available, while measuring the coupled velocity and pressure fields is more challenging. In such cases, we aim to predict the future evolution of the concentration field, as well as the velocity and pressure fields, ***based solely on concentration data***.
> > >
> > > Our 2D synthetic data consists of grayscale video sequences of the concentration field, where the physical scale is unknown. This essentially forms a sptiotemporal forecasting problem (e.g., predicting the evolution of the concentration field based on the history measurement), where models like Neural Operators (e.g., FNO, U-NO, LSM, etc.) can be used. Given the pure data-driven nature of the baselines, they cannot predict the underlying velocity or pressure fields, while couple with the concentration field, due to the lack of training labels for these quantities. In contrast, our method makes it possible to simultaneously estimate and predict the velocity and pressure fields.
> > >
> > > Regarding the quantitative comparisons, since the baseline models (e.g., FNO, F-FNO, U-NO, U-FNO, LSM, etc.) cannot predict the velocity or pressure fields, we only compare the metrics of the predicted concentration field, as shown in Table 2 (Section 4.1, page 6) in the paper. We can see that our model outperforms these baselines with notible margins. Similarly,  for the SEVIR radar dataset, our evaluation focuses on the prediction accuracy for Vertically Integrated Liquid (VIL) (Section 4.2, Table 3, page 9) commonly used in the nowcasting tasks.
> > >
> > > In addition, since the HelmFluid model is capable of predicting the velocity field (but not the pressure field), we considered it a relevant comparison for our method. The comparative results on the estimated latent physical quantities are shown in **Table A** below. We can see that our predition subpasses that of HelmFluid.
> > >
> > > **Table A: Correlation between Predicted and True Latent Physical Quantities.**
> > >
> > > | Latent Physical Quantities | Correlation (PINP) | Correlation (HelmFluid) |
> > > |----------------------------|---------------------|------------------------|
> > > | $u_x$                      | 0.9017              | 0.8891                 |
> > > | $u_y$                      | 0.9109              | 0.8997                 |
> > > | $p$                        | 0.8984              | -                      |

---

> ### Author Response · Authors · 2024-11-26
> **Reply to additional comment by Reviewer m8WU: clarification on the evaluation metrics**
>
> > **Comment 2: Clarification on the evaluation metrics.**
>
> **Reply:** We would like to clarify the rational use of correlation as the evaluation metric. As discussed in [1], correlation is a commonly used metric in evulating the similarity between two quantities. In particular, when the correlation exceeds 0.9, the variables is strongly correlated.
>
> We appreciate your suggestion on the use of MSE to evaluate the predicted velocity and pressure fields. We would like to highlight that due to **the unknown physical scale** (since only the grayscale video sequences of the concentration field are provided as the training data), our predicted velocity fields and the true values have a scaling relationship, expressed as follows:
>
> $$\hat{\boldsymbol{u}} = \alpha \boldsymbol{u}$$
>
> where $\alpha$ is a scaling constant related to the actual physical scale. When only concentration videos are available, we cannot determine the exact physical scale of the region represented in the video (e.g., the scale may lie in any range such as from 10 cm to 1 m in the smoke dataset). This uncertainty introduces a constant factor difference between the predicted and true values.
>
> Thus, directly computing the MSE might be inappropriate. To address this, we considered two methods for calibration:
>
> - Case 1: Using the true initial velocity values (assumed known *a priori*) to compute $\alpha$.
> - Case 2: Using just one true data point to estimate $\alpha$.
>
> We then apply the computed $\alpha$ to correct the predicted results (note that HelmFluid also requires calibration). The calibrated MSE values are shown in **Table B** below.
>
> **Table B: Comparison of calibrated MSE for latent physical quantities between our model and HelmFluid.**
>
> | Latent Physical Quantities | MSE: PINP (Case 1) |MSE: PINP (Case 2) | MSE: HelmFluid (Case 1) | MSE: HelmFluid (Case 2) |
> |----------------------------|---------------|---------------|--------------------|--------------------|
> | $u_x$                      | 0.1056        | 0.3128        | 0.1482             | 0.5035             |
> | $u_y$                      | 0.0806        | 0.2806        | 0.1187             | 0.3407             |
> | $p$                        | 0.1539        | 0.4653        | -                  | -                  |
>
> The results in Tables **A** and **B** highlight the superiority of our approach in predicting the latent physical quantities. Moreover, even with a constant factor difference, a high correlation (>0.9) is sufficient to demonstrate the consistency between our predictions and the true data in terms of distribution, regardless of scaling. This can serve as supplementary information for concentration prediction and provide interpretable evidence for the future evolution of fluid dynamics.
>
> In conclusion, with concentration data as the sole training labels, our model **outperforms** all baselines for all tested examples.
>
> Additionally, we have also revised Table 1 (page 3, Section2), with the modifications highlighted in blue to enhance the clarity of the paper (please see the **updated .pdf file**). Thank you very much for your insightful comments, and we look forward to your feedback!
>
>
> ***Reference:***
>
> [1] Akoglu, Haldun. "User's guide to correlation coefficients." *Turkish Journal of Emergency Medicine* 18.3 (2018): 91-93.

---

> > ### Comment · Reviewer_m8WU · 2024-11-26
> >
> > Thanks for your more detailed explanation.
> >
> > I understand the author's thoughts on the above two questions, but I still consider that this article has the overclaim for predicting velocity and pressure because, as the author mentioned, the obtained velocity and pressure lack a scaling constant. If we need to predict velocity and pressure, we often hope to obtain an accurate solution. The author needs to revise the manuscript to weaken the emphasis on predicting velocity and pressure except for improving interpretability, and at least Table 1 is unnecessary. From the results, it is acceptable to only predict concentration, so there is no need to emphasize velocity and pressure.
> >
> > Thanks to the author for providing the above two results, which are more meaningful than the qualitative images in the manuscript and should be included in the manuscript (main text or appendix). If the authors still hope to highlight the prediction of velocity and pressure, although I do not recommend that the author do so, at least Table A should be placed in the main text. Moreover, **I strongly recommend the author delete Table 1 and the image on the right, as it may lead to some confusion.**
> >
> > If the author is willing to revise the manuscript based on the above suggestions (both revisions are acceptable), I will reconsider my rating. Although there are still many issues with the manuscript, this work is meaningful, and most importantly, I hope to encourage and support the author's experiments on real-world data.

---

> > > ### Author Response · Authors · 2024-11-26
> > > **The paper has been revised**
> > >
> > > Thank you for your encouraging feedback!
> > >
> > > In accordance with the reviewer's suggestions, we have made revisions and adjustments to the paper (indicated in blue color). Please see **the updated .pdf file**. In the revised version, we have made the following revisions:
> > >
> > > - We have removed Table 1 and the image on the right.
> > >
> > > - We have weakened and  thoroughly revised our description of the predictions for velocity and pressure (including the abstract).
> > >
> > > - We have stated in the main text that scaling relationship exists between the estimated latent physical quantities and their true values (lines 395-401, Section4.1, page 8).
> > >
> > > - We have added the experiments comparing with Helmfluid, the explanation of calibrated MSE, and the two experimental results mentioned by the reviewer to Appendix D.
> > >
> > > Thank you very much for your great suggestion. Please let us know if you have any other questions. Looking forward to your response!

---

> > > > ### Comment · Reviewer_m8WU · 2024-11-27
> > > >
> > > > Thanks for your response. I raised my rating to 6.

---

> > > > > ### Author Response · Authors · 2024-11-27
> > > > > **Thanks for your positive feedback**
> > > > >
> > > > > Dear Reviewer m8WU,
> > > > >
> > > > > Thank you very much for your positive and encouraging feedback. Your constructive comments, as well as time and effort placed on reviewing our paper, are highly appreciated!
> > > > >
> > > > > Best regards,
> > > > >
> > > > > The Authors

---

### Official Review · Reviewer_zzB8 · 2024-11-01

**Soundness:** 2
**Presentation:** 3
**Contribution:** 2
**Rating:** 6
**Confidence:** 4

**Summary:**

This paper proposed the PINP method for fluid prediction. The method predicts future fluid fields by learning velocity and pressure simultaneously from partial observations. The authors employ a physical inference neural network to predict several physical quantities of the flow field at a particular moment. For the next timestep, they utilizes discrete PDEs predictor and correction network to generate the flow field. The training of the model is refined through the application of MSE loss, equation loss, and a temporal constraint loss. The proposed method shows advantages compared to several baselines on both synthetic and real-world data.

**Strengths:**

This paper combines equation loss with operator learning through the Navier-Stokes equation, integrating physics-driven and data-driven approaches by learning unobserved physical quantities.
- An innovative point of the proposed method is incorporating the equation loss commonly used in PINN, including incompressible Navier-Stokes equations, into predicting potential physical quantities of the future flow field.
- The figures provided in the paper can accurately reflect the characteristics of the model.
- The ablation of the paper accurately explains the role played by each module.

**Weaknesses:**

This paper has some weaknesses, including:
- Some datasets are too simple or may theoretically not match the methods to some extent.
  - The fluid motion in the Flow 2D dataset is relatively slow, and the dynamics are not as complex as those encountered in more advanced fluid dynamics scenarios.
  - The fluids in real datasets may not strictly adhere to the incompressible Navier-Stokes equations. Consequently, the physical constraints proposed in this paper might encounter limitations when applied to more diverse or complex fluid systems.
- The improvements observed in specific datasets, such as Smoke3D, are modest. For example, the MAE and MSE metrics of Smoke 3D only demonstrate a marginal enhancement.
- The paper's grammar and expression could be improved. In some instances, the clarity of the writing detracts from the overall quality of the paper, potentially hindering the reader's understanding of the research.

Specific issues and possible improvements will be discussed in the next section.

**Questions:**

- For the sole real dataset, SEVIR, raises several concerns:
  - It is unclear why the MSE metric for the proposed method underperforms compared to most other baselines. A detailed analysis and discussion of this discrepancy would be beneficial.
  - Given the potential phase transition of water in the SEVIR dataset, it is questionable whether the fluid satisfies the incompressible property. The authors are encouraged to discuss the applicability of the equation loss function to real datasets with such characteristics.
  - I highly recommend the authors compare with the traditional numerical method pySTEPS[1], which predicts future fluid fields by estimating potential velocity fields and extrapolating optical flow. This method has the ability to accurately estimate extreme values.
- The paper does not specify the fluid's Reynolds number, which is crucial for understanding the flow characteristics. The fluid in the Fluid 2D dataset appears to represent simple laminar flows. The authors are recommended to provide experiments with more turbulent datasets to enhance the paper's practical value.
- The paper lacks information on how the baselines were trained, and some baselines exhibit abnormal flickering. The authors should recheck the training process for all baselines or explain these anomalies.
- The visualization of velocity in the paper is not as intuitive as it could be. Utilizing tools such as *pyplot.quiver* in *matplotlib* to depict the velocity field based on flow field observations as supplementary material is suggested.
- The grammar and expression throughout the paper should be improved. A thorough review and enhancement are advised.


[1] https://pysteps.github.io/

---

> ### Author Response · Authors · 2024-11-21
> **Reply to Reviewer zzB8 (Part 1)**
>
> Thank you for your constructive comments and suggestions!
>
> > **Q1. Concerns on SEVIR: (1) underperformed MSE metric, (2) adaptability of NS equations in nowcasting, (3) comparison with PySteps.**
>
> **Reply**: Insightful comments! We would like to firstly clarify that, although the Navier-Stokes (NS) equations are not strictly satisfied in SEVIR, the corresponding test was taken to evaluate our model's predictive performance and robustness to noise under conditions where the equations are not fully obeyed. Importantly, no specific adaptations, such as adjustments to the loss function weights, were designed for this task. As noted, the nowcasting scenario may not strictly adhere to the NS equations. However, we believe that incorporating the NS equations helps to capture the fundamental motion characteristics of nowcasting.
>
> To address the first two points raised, we conducted the following experiments:
>
> - Lowered the physical loss weight (weight = 0.1) while keeping other settings unchanged.
> - Removed the physical loss entirely.
>
> The results are summarized below in **Table A**.
>
> **Table A: Comparison of Evaluation Metrics on the SEVIR dataset under Different Physical Loss Configurations.**
> |Experiments | MSE(1e-3) | CSI-M | CSI-219 | CSI-181 | CSI-160 | CSI-133 | CSI-74 | CSI-16 |
> |--------------------------------|-----------|-------|---------|---------|---------|---------|-------|-------|
> | Lowered Physical Loss Weight   | 4.0305    | 0.3662| 0.0781  | 0.1724  | 0.2213  | 0.3402  | **0.6563**| 0.7288|
> | No Physical Loss | **4.0092**    | 0.3569| 0.0674  | 0.1541  | 0.2081  | 0.3274  | 0.6547| **0.7299**|
> | PINP (Our Work) | 4.1684    | **0.3800**| **0.0989**  | **0.1952**  | **0.2448**  | **0.3573** | 0.6556| 0.7279|
>
> The results in **Table A** demonstrate that while nowcasting processes do not strictly adhere to the NS equations, these equations effectively capture certain underlying motion characteristics. When the physical loss weight is reduced, loosening the constraints, the Mean Squared Error (MSE) decreases. However, this relaxation comes at the cost of reduced ability to model essential motion features, as reflected in the decline of the Critical Success Index (CSI). The corresponding analysis and results have been included in Appendix D.4 (page 20).
>
> Regarding comparisons with pySTEPS, we sincerely appreciate the reviewers' suggestion. However, after careful consideration, we believe this is unnecessary since NowcastNet (our comparable baseline model) has already demonstrated improvements over pySTEPS.
>
>
> > **Q2. Simplicity of the Fluid 2D Dataset.**
>
> **Reply**: When calculating the Reynolds number, we defined the characteristic length as the length of the widest obstacle, resulting in the Reynolds number around 450. For the 2D fluid training set, the obstacle positions were varied. However, in the test set, we increased complexity by altering obstacle shapes and numbers in addition to their positions. This introduces greater variability and poses higher demands on the neural network's ability to learn motion laws and generalize effectively.
>
>
> > **Q3. Flickering in Baseline Predictions.**
>
> **Reply**: The flickering phenomenon observed in some baseline model predictions is a form of periodic oscillation. Initially, this puzzled us as we did not find any bug in the implementation code. Upon further investigation, we discovered that the oscillation period (4 steps) matches the training rollout step size. Specifically, our training uses 4 frames to predict one frame unrolling over four steps, while the testing (inference) extends the rollout process to 40 steps. This periodic oscillation may arise because some models (except ours) struggle to learn motion dynamics within short sequence training.
>
> To further investigate this phenomenon, we conducted three validation experiments as follows:
>
> - Case 1: Training with four frames predicting one frame for four steps.
> - Case 2: The same as Case 1 but with an adjusted batch size.
> - Case 3: Training with four frames predicting one frame for six steps.
>
> Our findings include:
>
> - Experiments on Case 1 and Case 2 exhibited oscillations with a period of 4 throughout training (even at epoch 200).
> - Experiment on Case 3 initially had oscillations with a period of 6, but these gradually diminished as training progressed.
>
> This suggests that some baseline models struggle to capture long-term motion features when trained on short-term sequences. Under the configuration of Case 3, the oscillations in LSM were significantly reduced, yet our model still outperformed it, as shown in **Table B** below.
>
> **Table B: Comparison of Results Between Our Method and LSM Under Experimental Setting Case 3.**
> | Method      | MAE   | MSE    |
> |-------------|-------|--------|
> | LSM         | 0.0099| 0.0002 |
> | PINP (ours) | 0.0062| 0.0001 |
>
> The details of the above analysis and results are included in our paper (page 7, Section 4.1 and page 29, Appendix F).

---

> ### Author Response · Authors · 2024-11-21
> **Reply to Reviewer zzB8 (Part 2)**
>
> > **Q4. Velocity Visualization.**
>
> **Reply**: Excellent suggestion! While we believe the field-based visualization provides more comprehensive information compared with the arrow-based visualization, we understand its limitation in intuitiveness. Therefore, followed your suggestion, we have included the arrow-based visualization in Figure S10 (page 22, Appendix E) for better illustration.
>
>
> > **Q5. Writing and Grammar.**
>
> **Reply**: We appreciate this feedback. We have thoroughly revised the paper to improve the language and formatting, ensuring clearer and more effective presentation. Please see our revisions marked in red color in the revised paper.
>
>
> > **Weakness: Modest Improvements on Smoke3D.**
>
> **Reply**: We acknowledge the modest improvements observed on specific datasets like Smoke3D. However, we achieved consistently the best results across all simulation datasets. Additionally, we would like to emphasize that our contributions go beyond the prediction accuracy. Our model is capable of predicting hidden physical quantities (e.g., velocity and pressure) in an unsupervised manner. These predicted quantities align closely with real physical fields and serve as interpretable evidence which deepen the understanding of future motion processes. This enhances the value of our approach. Hope this clarifies your concern.
>
> In summary, we appreciate your constructive comments and suggestions. Please let us know if you have any other questions. We look forward to your feedback!

---

> ### Comment · Reviewer_zzB8 · 2024-11-21
> **Thank you for your response.**
>
> Thank you for the extra experiments and detailed answers.
>
> Q1. Concerns on SEVIR:
> Firstly, I would like to express my gratitude to the authors for explaining the applicability of the N-S equation as a loss function. I think this helped the model learn better physical characteristics to a certain extent. However, this paper does not include visualization of the physical quantities learned from the SEVIR dataset. I would be very interested in looking at the visualization of the learned velocity fields using pyplot.quiver and comparing it with the velocity fields learned by NowcastNet.
>
> Secondly, I appreciate the authors' explanation for the MSE metric, but I still believe this indicates that the authors have not treated the baseline fairly. As shown in Fig. 8, the predicted intensity of PINP is significantly higher than that of other models. This does not clarify whether the higher CSI compared to other models is due to higher forecast intensity by introducing various losses or the more accurate position provided by velocity prediction.
>
> Thirdly, the authors stated that only the Evolution Network part was used for NowcastNet. This network applies an advection scheme directly on the input observation. Thus, the results should not show a decrease in intensity, which contradicts the results provided in Figures S17 and S16. Additionally, this does not demonstrate that the Evolution Network part has already shown improvements over pySTEPS. As mentioned in the previous point, this severely affects the fairness of the CSI metric comparison. Therefore, I insist on suggesting that the authors check the correctness of the implementation of the Evolution Network or provide comparisons with pySTEPS.
>
> Q2. About Fluid 2D dataset:
> I appreciate the authors' provision of the parameters utilized within the Fluid 2D Dataset. However, the motion characteristics of visualization suggest no turbulence presence in the dataset. I highly recommend that the authors consider employing a dataset with a higher Reynolds number that encompasses turbulence or leverage the Bounded N-S dataset from the baseline HelmFluid to ascertain the model's efficacy in forecasting turbulent flows. Otherwise, the practical application value of this model in real fluid prediction scenarios is somewhat limited.
>
> Q3. Flickering in Baseline Predictions.
> Thank you to the authors for explaining the prediction results' flickering and providing additional training outcomes. I have no further questions about this.
>
> Q4. Velocity Visualization.
> I appreciate the authors' visualization of the velocity field. The current results are clear and well-presented.
>
> Q5. Writing and Grammar.
> Thank you for revising the paper and updating the expressions. Since the entire article is now more readable, I have increased the presentation score to 3.
>
> Weakness: Modest Improvements on Smoke3D.
> I also appreciate the authors' explanation regarding the issues on the Smoke3D dataset. However, even though the model provides additional variable predictions with physical interpretability, I still believe that the improvement in the metrics is not significant enough.
>
> In summary, the authors have partially addressed my concerns about this paper. However, (1) the issue with the baseline and metrics on the SEVIR dataset still exists, and (2) the problem that the Fluid 2D Dataset only contains laminar flow and is overly simplistic remains unresolved. Therefore, I will not change my score for now.

---

> > ### Author Response · Authors · 2024-11-25
> > **Request your feedback before the end of the discussion period**
> >
> > Dear Reviewer zzB8:
> >
> > As the author-reviewer discussion period will end soon, we would appreciate it if you could review our responses at your earliest convenience. If there are any further questions or comments, we will do our best to address them before the discussion period ends.
> >
> > Thank you very much for your time and efforts. Looking forward to your response!
> >
> > Sincerely,
> >
> > The Authors

---

> > > ### Comment · Reviewer_zzB8 · 2024-11-25
> > >
> > > Thank you to the authors for adding ablations and more experiments.
> > >
> > > My concerns are mostly resolved. However, I have stated that no turbulent flow exists according to the videos on the Fluid 2D dataset. No vortex is formed near the flow around the cylinder where turbulence should occur, which cannot reflect the prediction ability of this method for turbulence.
> > >
> > > Considering the above, I have increased my score to 5.

---

> > > > ### Author Response · Authors · 2024-11-25
> > > > **Clarification on Reviewer zzB8's Question**
> > > >
> > > > Thanks for your additional feedback.
> > > >
> > > > > **Q1: Concerns regarding the absence of turbulence in the Fluid 2D dataset.**
> > > >
> > > > **Reply**: Thank you for your feedback. Referring to [1] (Table 1.1, page 22), the flow regime is determined by the Reynolds number. Specifically:
> > > >
> > > > - Case 1: When the Reynolds number lies between $300$ and $1.3 \times 10^5$, the flow is classified as subcritical, characterized by vortex street instability.
> > > >
> > > > - Case 2: Complete turbulent separation occurs when the Reynolds number exceeds $3.5 \times 10^6$.
> > > >
> > > > Our Fluid 2D dataset ($Re$ = 450, 1800, 2700) corresponds to the Reynolds number range in Case 1, where the vortex street instability possesses some extent of turbulence characteristics. While the Fluid 2D dataset does not reach the full turbulent regime, the SEVIR dataset involves Reynolds numbers within the range in Case 2, making it appropriate for evaluating turbulence-related phenomena, including turbulent separation. We hope this clarifies the suitability of the datasets for different flow regimes.
> > > >
> > > > In addition, we would like to clarify the primary problem we are motivated to solve as follows:
> > > >
> > > > "*Our goal is develop a generalizable model to predict concentration of a scalar field (e.g., smoke, dye, precipitation, etc.) while simultaneously estimating velocity and pressure fields (without any training labels) as interpretable evidence in an unsupervised manner.*"
> > > >
> > > > Thank you very much for your great questions. Please let us know if you have any other questions. Looking forward to your response!
> > > >
> > > > ***Reference:***
> > > >
> > > > [1] Schlichting, Hermann, and Klaus Gersten. Boundary-layer theory. springer, 2016.

---

> > > > > ### Comment · Reviewer_zzB8 · 2024-11-26
> > > > >
> > > > > Thank you for your explanations. Although the provided video does not show the Karman vortex street phenomenon, the authors added additional experiments to show the model's performance on turbulent flow. I decided to raise my score to 6.
> > > > >
> > > > > Additionally, it should be pointed out that the presence of Karman vortex streets does not necessarily mean that the flow field is turbulent.

---

> > > > > > ### Author Response · Authors · 2024-11-26
> > > > > > **Thanks for your positive feedback**
> > > > > >
> > > > > > Dear Reviewer zzB8,
> > > > > >
> > > > > > Thank you very much for your positive and encouraging feedback. Your constructive comments, as well as time and effort placed on reviewing our paper, are highly appreciated!
> > > > > >
> > > > > > Best regards,
> > > > > >
> > > > > > The Authors

---

> ### Author Response · Authors · 2024-11-22
> **Reply to your additional comments**
>
> Thanks for your additional comments and suggestions!
>
> > **Q1a: Velocity Field Visualization and Comparison with NowcastNet.**
>
> **Reply**: Excellent suggestion! We have included the figure comparing the velocity fields of our method with those of NowcastNet and pySTEPS in Figure S11 (Section E, page 23) in the revised paper.
>
> > **Q1b: The comparison with pySTEPS**
>
> **Reply**: Great suggestion! We have run pySTEPS, obtained the prediction results, and compared this method with NowcastNet (under different settings) as well as our own method. The comparison results are shown in **Table A** below, and the above results have already been updated in our revised paper (Table S1, page 16, Section A.5).
>
> **Table A: Comparison of pySTEPS with NowcastNet (under different settings) as well as our own method.**
> | Experiment                           | MSE(1e-3) | CSI-M  | CSI-219 | CSI-181 | CSI-160 | CSI-133 | CSI-74  | CSI-16  |
> |--------------------------------------|-----------|--------|---------|---------|---------|---------|---------|---------|
> |pySTEPS|8.4682|0.3373|0.0912|0.1756|0.2162|0.3244|0.5848|0.6313|
> | Evolution Network (wrap mode='bilinear') | **3.8883**  | 0.3743 | 0.0803  | 0.1811  | 0.2351 | 0.3515 | **0.6635** | **0.7339** |
> | Evolution Network (wrap mode='nearest')  | 6.1783    | 0.3529 | 0.0971 | 0.1858 | 0.2220  | 0.3110  | 0.5986  | 0.7027  |
> | NowcastNet                           | 5.7999    | 0.3549 | 0.0933  | 0.1844  | 0.2179  | 0.3104  | 0.6131  | 0.7106  |
> | PINP (our work)                      | 4.1684  | **0.3800** | **0.0989** | **0.1952** | **0.2448** | **0.3573** | 0.6556  | 0.7279  |
>
> Based on the results shown in **Table A**, our method outperforms pySTEPS.
>
> > **Q1c: The issue of weakened prediction intensity in NowcastNet.**
>
> **Reply**: Excellent remark! This weakening is caused by the "bilinear" mode used during "wrap". If the "nearest" mode is applied, the weakening issue does not occur.
>
> As shown in **Table A**, the reason for choosing "bilinear" mode is that it performs relatively better on the evaluation metrics. The new comparison results are added and presented in our revised paper (Table S1, page 16, Section A.5).
>
> > **Q2: Evaluation on more complex fluid datasets.**
>
> **Reply**: We understand the reviewer’s concern. We generated data with incident velocities 4 times and 6 times the original velocity (corresponding to Reynolds numbers of 1800 and 2700) and tested our trained model (inference). The results are shown in **Table B** below. Our model still achieves the best performance.
>
> **Table B: Results at higher Reynolds numbers.**
>
> |Method|MSE|
> |--|--|
> |PINP|**0.0157**|
> |HelmFluid|0.0807|
> |LSM|0.0364|
> |FNO|0.0226|
> |U-NO|0.0226|
> |U-FNO|0.0181|
>
> Moreover, for your suggestion regarding testing the Bounded N-S dataset used in Helmfluid, we feel a little bit confused and believe it might not be suitable for evaluation of our model on turbulence prediction, as its Reynolds number is only 300 (Table 8, Appendix B.1 in HelmFluid). Anyway, thank you for your suggestions!
>
> **Concluding Remark:** In summary, we appreciate your constructive comments and suggestions. Please let us know if you have any other questions. We look forward to your feedback!

---

### Author Response · Authors · 2024-11-21
**General reply**

Dear Reviewers:

We would like to thank you for your constructive comments, which are very helpful in improving our paper. We have posted the point-to-point replies to each question/comment raised by you and submitted our revised paper.

We are pleased that the reviewers recognized our work. In particular, we thank the reviewers for recognizing the ***impressive results*** (m8WU, tKs8, zzB8), ***thorough experiments*** (m8WU, 3QEU) and ***well-written paper*** (3QEU, tKs8).

Comprehensive revisions and adjustments (indicated in red color) have also been made in the revised paper (please see **the updated .pdf file**). In the revised version, we have made the following revisions:

- Added experiments and analysis on the issue of anomalous flickering in Section 4.1 (page 7) and Appendix F (page 29).

- Added new ablation experiments: Model performance in predictions without Velocity-Pressure constraints on Table 4 (page 9, Section 5) and Appendix D.1 (page 19).

- Added details on the experimental setup and result for NowcastNet (Line 429-431, page 8, Section 4.2 and Appendix A.5, page 15).

- Added an analysis of the impact of physical loss on the SEVIR dataset (Appendix D.4, page 21).

- Added Table 4 (page 9, Section 5) to the main text, which includes all our ablation experiment results  and redrew Figure 9 (page 10, Section 5).

- Added the correlation between the predicted latent physical quantities and the true values as a quantitative result (Figure 6\(c\), page 8, Section 4.1).

- Added a new traditional numerical method, pySTEPS, for comparison (Table S1, page 16, Section A.5) in  nowcasting.

- Revised the temporal expressions throughout the paper.

- Thoroughly revised the paper, including formatting and textual expressions.

Please do feel free to let us know if you have any further questions.

Thank you very much.

Best regards,

The Authors of the Paper

---

### Author Response · Authors · 2024-12-03
**Special thanks to all reviewers**

Dear Reviewers,

We would like to express our sincere gratitude for the time and effort you dedicated to reviewing our paper. Your thoughtful comments and constructive suggestions have been extremely valuable in improving the quality of our work. The discussions prompted by your feedback have been both productive and insightful, significantly enhancing the overall clarity of our paper.

Especially in the explanation of the baseline model (by Reviewer zzB8 and 3QEU), the comparison with pySTEPS (by Reviewer zzB8), further clarification on the validity of the pressure field $p$ (by Reviewer 3QEU), additional clarification of the estimation of latent physical quantities (by Reviewer m8WU), and various improvements in expression (by Reviewer m8WU and 3QEU), your suggestions have been extremely helpful. They have greatly enhanced the presentation of our research, making it more comprehensive and rigorous. We deeply appreciate your thorough review and constructive comments, which have helped us refine our methodology and ensure the reliability of our findings. Additionally, we would like to thank the Reviewer tKs8. Your recognition of our work is a tremendous encouragement to us.

Once again, thank you all for your invaluable contributions to this work!

Best regards,

The Authors

---

### Meta-Review · Area_Chair_ZX8P · 2024-12-20

**Metareview:**

This paper introduces a physics-informed neural predictor for fluid dynamics. The approach consists of a physical inference network, which estimates latent physical quantities, with a discrete PDE-based predictor that generates fluid flows, leading to improved accuracy and interpretability. The method is evaluated through extensive experiments on both synthetic and real-world data. While the design is generally considered reasonable, some reviewers criticized it as overly complex and not entirely straightforward, making the paper harder to follow. Nonetheless, the reviewers recognized the method’s sound rationale and superior performance. Given the positive reviews and very engaged discussions amongst authors and reviewers, I recommend accepting this paper.

**Additional Comments On Reviewer Discussion:**

Reviewer zzB8 was initially concerned about the benchmarks and baselines, but these issues were effectively addressed through additional experiments presented during the rebuttal. Both zzB8 and m8WU highlighted unclear writing, which led to confusion regarding the derivations, architecture, evaluation metrics, etc. Reviewer 3QEU raised concerns about the overclaimed initial title, which is revised to be more specific, as well as the rationality behind the method design, which is also clarified with in-depth discussion.

---

### Decision · Program_Chairs · 2025-01-22

Accept (Poster)